# Single-cell profiling reveals periventricular CD56[bright] NK cell accumulation in multiple sclerosis

**Sabela Rodríguez-Lorenzo[1†], Lynn van Olst[1†], Carla Rodriguez-Mogeda[1], Alwin Kamermans[1], Susanne MA van der Pol[1], Ernesto Rodríguez[1,2], Gijs Kooij[1], Helga E de Vries[1]\***

[1]MS Center Amsterdam, Department of Molecular Cell Biology and Immunology, Amsterdam Neuroscience, Vrije Universiteit Amsterdam, Amsterdam UMC, Amsterdam, Netherlands; [2]Amsterdam UMC, Vrije Universiteit Amsterdam, Department of Molecular Cell Biology and Immunology, Cancer Center Amsterdam, Amsterdam Infection and Immunity Institute, Amsterdam, Netherlands

**\*For correspondence:**
he.devries@amsterdamumc.nl

[†]These authors contributed equally to this work

**Competing interest:** The authors declare that no competing interests exist.

**Abstract** Multiple sclerosis (MS) is a chronic demyelinating disease characterised by immune cell infiltration resulting in lesions that preferentially affect periventricular areas of the brain. Despite research efforts to define the role of various immune cells in MS pathogenesis, the focus has been on a few immune cell populations while full-spectrum analysis, encompassing others such as natural killer (NK) cells, has not been performed. Here, we used single-cell mass cytometry (CyTOF) to profile the immune landscape of brain periventricular areas – septum and choroid plexus – and of the circulation from donors with MS, dementia and controls without neurological disease. Using a 37-marker panel, we revealed the infiltration of T cells and antibody-secreting cells in periventricular brain regions and identified a novel NK cell signature specific to MS. CD56[bright] NK cells were accumulated in the septum of MS donors and displayed an activated and migratory phenotype, similar to that of CD56[bright] NK cells in the circulation. We validated this signature by multiplex immunohistochemistry and found that the number of NK cells with high expression of granzyme K, typical of the CD56[bright] subset, was increased in both periventricular lesions and the choroid plexus of donors with MS. Together, our multi-tissue single-cell data shows that CD56[bright] NK cells accumulate in the periventricular brain regions of MS patients, bringing NK cells back to the spotlight of MS pathology.

## Editor's evaluation

This is a well-written, well-illustrated and well-conducted study of the immune cell landscape of multiple sclerosis (MS) tissue, with a particular focus on the periventricular region (septum) and choroid plexus, using single-cell mass cytometry (CyTOF). Overall the work is an impressive analysis of an understudied cell type in MS and represents an important resource.

## Introduction

Multiple sclerosis (MS) is a chronic neuroinflammatory disease characterised by demyelinating lesions within the central nervous system (CNS). In MS, peripheral immune cells gain access to the CNS and cause severe neuroinflammation, myelin damage, and subsequent neurodegeneration. In the past decades, a wealth of knowledge has been gained on the role of monocyte-derived macrophages, CD8[+] and CD4[+] T cells, B cells and antibody-secreting cells in MS pathogenesis (***Machado-Santos***

*et al., 2018*; *Vogel et al., 2013*). To date, however, we still know relatively little about the presence and roles of other immune cell subsets in the MS brain, such as natural killer (NK) cells.

Brain regions around the ventricles are hotspots for MS lesions (*Brownell and Hughes, 1962*; *Jehna et al., 2015*; *Palmer et al., 1999*; *Simon et al., 1986*), but underlying mechanisms are poorly understood (*Pardini et al., 2021*). Since the majority of periventricular MS lesions occur around a central vessel (*Adams et al., 1987*; *Tallantyre et al., 2011*), it has been suggested that vascular topography may influence MS pathology (*Martinez Sosa and Smith, 2017*). Although the accumulation of immune cell infiltrates around post-capillary venules suggests their trafficking across the blood-brain barrier (*Smolders et al., 2020*), periventricular veins drain to the cerebrospinal fluid (CSF) (*Pardini et al., 2016*). Thus, MS periventricular pathology may be related to factors from both the blood and the CSF.

In MS, the CSF that flows through the ventricles is enriched in immune cells (*Cepok et al., 2001*; *Rodríguez-Martín et al., 2015*; *Schafflick et al., 2020*) and inflammatory factors (*Khaibullin et al., 2017*; *Vidaurre et al., 2014*). The main source of CSF is the choroid plexus, a secretory tissue located in the brain ventricles that acts as an immunological hub (*Strominger et al., 2018*) and forms the blood-CSF barrier. Thus, the location and functions of the choroid plexus are strategic to regulate periventricular homeostasis and thereby influence neuroinflammation (*Ghersi-Egea et al., 2018*; *Monaco et al., 2020*). For example, pro-inflammatory immune cells infiltrate the brain through the choroid plexus in the early stages in an MS animal model (*Reboldi et al., 2009*). On the contrary, in progressive MS patients, the choroid plexus may be a source of neuroprotective factors in response to chronic periventricular damage (*Rodríguez-Lorenzo et al., 2020a*). Moreover, we and others have shown that immune cells accumulate at the choroid plexus in the progressive phases of the disease (*Rodríguez-Lorenzo et al., 2020b*; *Vercellino et al., 2008*). We postulate that CSF-mediated immune processes originating in the choroid plexus participate in the periventricular inflammation typical for MS patients. However, it is still uncertain whether such processes involve cell trafficking into the CNS and/or the secretion of pro- or antiinflammatory factors.

Here, we used single-cell mass cytometry (CyTOF) to profile the immune landscape of periventricular MS brain regions and to better understand their pathology and routes of immune cell infiltration into the CNS. For this purpose, we isolated immune cells from the human postmortem periventricular brain areas (septum), the choroid plexus and the blood. With our 37-marker panel, we defined the main innate and adaptive immune cell populations and their phenotype in MS, dementia, and control donors without neurological disease. Besides detecting the accumulation of T cells and antibody-secreting cells typical of MS in the periventricular brain regions, we also identified an NK signature consisting of CD56[bright] NK cells. CD56[bright] NK cells are known for their immunoregulatory properties and were also detected in the blood of the same MS donors with a migratory and activated phenotype, similarly to those in the brain. Using multispectral immunofluorescence, we validated these findings in an independent cohort of periventricular brain and choroid plexus tissue, which indicated that NK cells expressing granzyme K, typical of the CD56[bright] subset, are enriched in MS lesions and the choroidal tissue from MS donors. Together, our study brings NK cells back to the spotlight of MS pathology by suggesting a local immunoregulatory role of the CD56[bright] subset within periventricular MS brain regions.

## Results

### Single-cell mass cytometry of the septum reveals an accumulation of T cells and natural killer cells in multiple sclerosis

To investigate the periventricular immune landscape in MS, we performed cytometry by time of flight (CyTOF) on postmortem septum – a periventricular brain region highly exposed to CSF –, choroid plexus – the main producer of CSF –, and peripheral blood from controls without neurological disease, donors diagnosed with dementia (used as a neurological control) and MS donors (*Figure 1a*). Demographic and clinical information of the patients is summarised in *Supplementary file 1* and *Figure 1—figure supplement 1*. We used a 37-antibody panel consisting of lineage markers, focused on lymphoid subsets, and phenotypic markers to determine the migratory, activation and memory phenotypes (*Supplementary file 2*). Differences in staining intensity between samples were minimal (*Figure 1—figure supplement 2*). We performed unsupervised clustering of the septum-derived cells

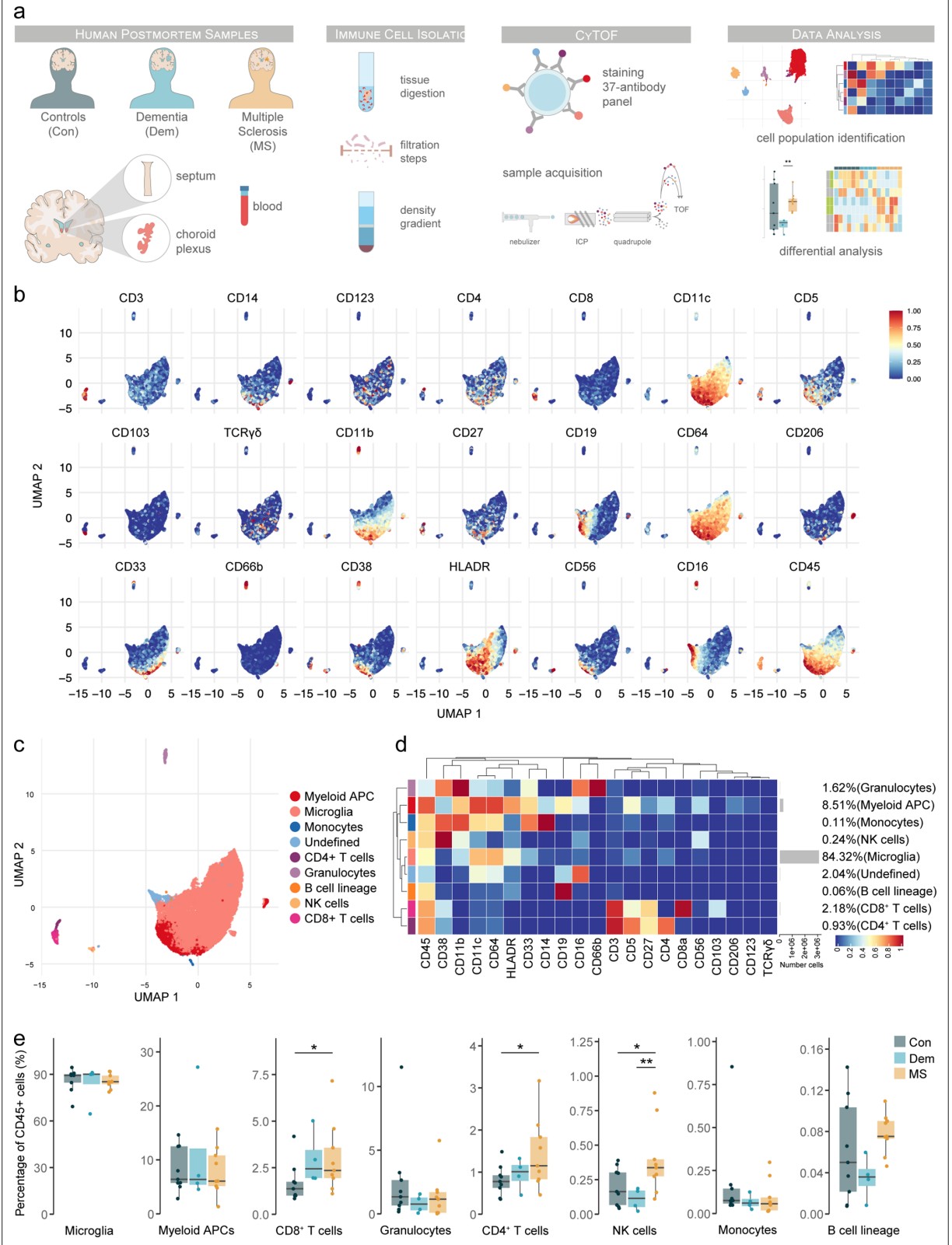

**Figure 1.** Immune phenotyping of the septum using mass cytometry reveals inflammation in multiple sclerosis involving T cells and natural killer cells. (**a**) Schematic overview of the study design. Immune cells were isolated from septum, choroid plexus and blood samples. Cells were then fixed, barcoded, stained and acquired on a CyTOF Helios system and analysed using FlowSOM and ConsensusClusterPlus. (**b**) UMAP plots are based on the arcsinh-transformed expression of the 'type' (lineage) markers in the septum-derived immune cells. A subset of 1000 randomly selected cells per

*Figure 1 continued on next page*

*Figure 1 continued*

sample is shown, colored according to the expression level of each marker. (**c**) UMAP plot based on the arcsinh-transformed expression of the 'type' (lineage) markers in the septum-derived immune cells. A subset of 1000 randomly selected cells is shown per sample, colored according to the manually annotated clusters. (**d**) Median scaled intensities of the 'type' (lineage) markers across the nine annotated septum-derived immune cell populations. The horizontal gray bars show the percentage of each cluster out of the total number of cells. (**e**) Percentage of each annotated cell population out of the total CD45+ cells from the septum of controls (Con), dementia (Dem) and multiple sclerosis (MS) donors. * adjusted p<0.1; ** adjusted p<0.05. Source data is listed in ***Source data 1***.

The online version of this article includes the following figure supplement(s) for figure 1:

**Figure supplement 1.** Demographic and clinical information.

**Figure supplement 2.** Staining intensity between the different barcoded CyTOF runs displayed using a reference sample.

**Figure supplement 3.** Immune phenotyping of the septum using mass cytometry reveals inflammation in multiple sclerosis involving T cells and natural killer cells.

**Figure supplement 4.** Myeloid subclustering in the septum.

**Figure supplement 5.** Correlations between the percentage of immune cells from the septum and clinical parameters.

followed by manual merging of the clusters based on biological knowledge (***Figure 1—figure supplement 3***).

Overall, in the septum, we identified 9 immune cell populations (***Figure 1b–d***) and one cluster of non-immune cells of possible neuronal origin (CD45- CD56bright) (***Figure 1—figure supplement 3d***), which was excluded from further analysis. The septum was mainly populated by microglia (84.3% of CD45+ immune cells), followed by other antigen-presenting cells (APCs) of myeloid origin (8.5 %). APCs of myeloid origin consisted of 3 subsets (80% were CD11chigh CD49d+ CD206-, 12.8% were CD11clow CD49d+ CD206+; and 7.3% were CD11chigh CD49d- CD206-) (***Figure 1—figure supplement 4***). The frequency of CD8+ T cells (2.2 %) more than doubled that of CD4+ T cells (0.9 %). Other immune cells present in the septum were granulocytes, natural killer (NK) cells, monocytes, and cells from the B lymphocyte lineage.

We next compared the proportions of each population in the septum among the MS and control groups (***Figure 1e***). We found a higher percentage of both CD8+ and CD4+ T cells in MS compared to controls. CD4+ T cell accumulation was specific for MS, while the percentage of CD8+ T cells was higher in both MS and dementia compared to controls. We also found a higher percentage of NK cells in the MS group compared to the control and dementia cases. A moderate negative correlation was found between the percentage of CD8+ T cells and age, while there was a moderate positive correlation between the B cell lineage frequencies and postmortem delay (PMD) (***Figure 1—figure supplement 5***).

Thus, we were able to identify the canonical brain immune cell populations in lesion-prone, periventricular brain regions using CyTOF. We found that the septum also presents an accumulation of CD8+ and CD4+ T cells typical of MS brains (***Babbe et al., 2000***; ***Machado-Santos et al., 2018***). Moreover, we show that NK cell accumulation is a signature of the septum of MS patients.

## Phenotyping the T cell populations in the septum

Next, we investigated which subsets within CD8+ and CD4+ T cells were accumulating in the septum of MS donors. Briefly, CD8+ and CD4+ T cells were further subdivided into 10 populations and a tentative biological name was assigned to each of them (***Figure 2a***, ***Figure 2—figure supplement 1***). The septum was mostly populated by CD69+ tissue-resident memory T cells (T$_{RM}$), belonging to both CD8+ (clusters CD8 c1 and CD8 c2, the latter expressing residency marker CD103) and CD4+ T cells (cluster CD4 c1). Effector memory T cells (T$_{EM}$) were also present (clusters CD8 c4 and CD4 c2), including a cluster of CD8+ T cells re-expressing CD45RA (T$_{EMRA}$ cells, cluster CD8 c3). A small cluster of CD8+ T cells presented an intermediate or transitional phenotype between naïve and memory (cluster CD8 c5). Naïve CD4+ T cells (cluster CD4 c3) and CD4+ T$_{REGS}$ (cluster CD4 c4) were present in small proportions. We also identified a small cluster of γδ T cells, which was not detected in the general clustering of the septum.

Proportions of the different T cell subsets present in the septum did not differ among the disease groups (***Figure 2b***). However, before correcting for multiple testing, the percentage of CD4+ T$_{EM}$ cells was higher in MS relative to dementia (unadjusted p=0.03), while the proportion of CD8+ T$_{EMRA}$ cells

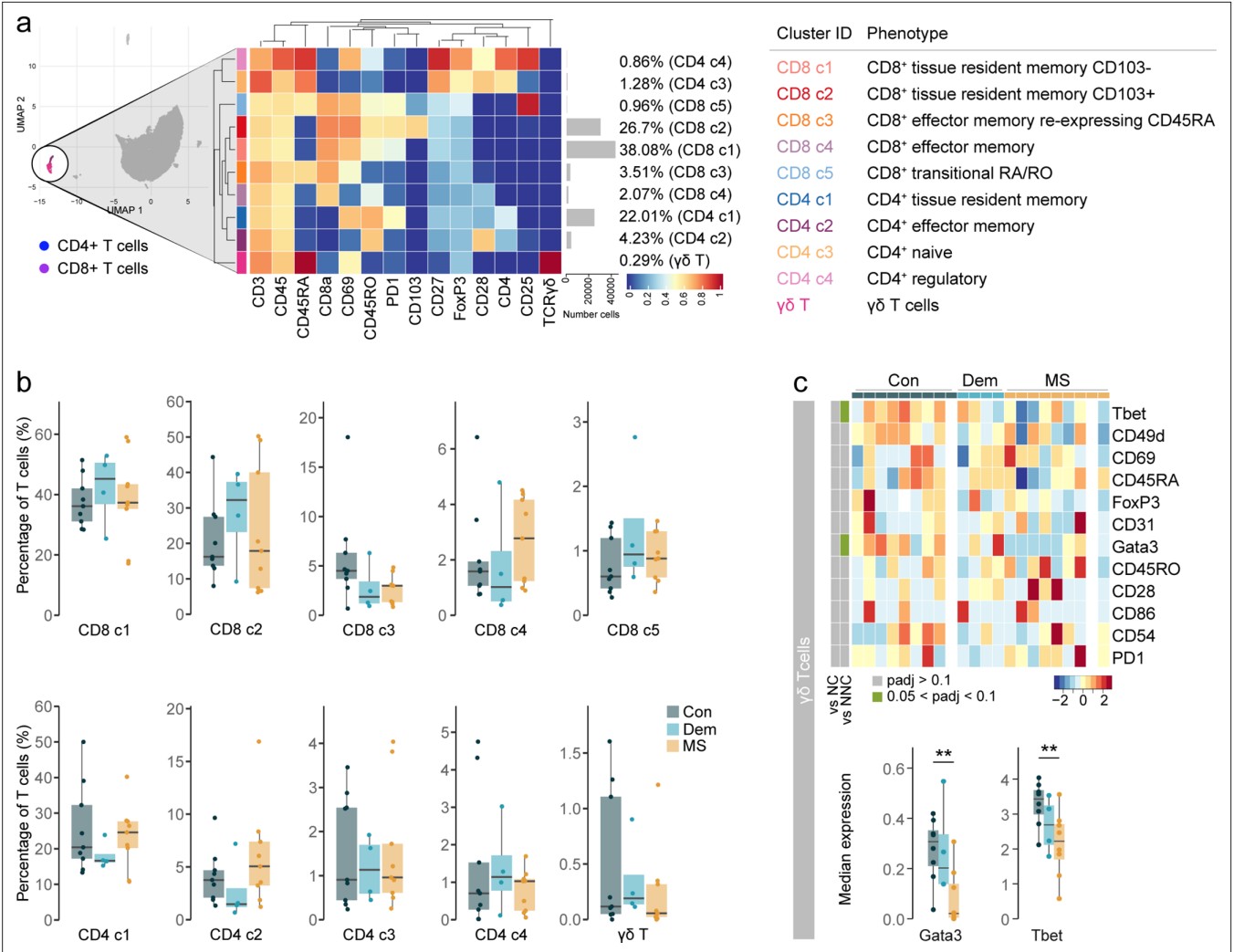

**Figure 2.** Phenotyping the T cell populations in the septum using mass cytometry. (**a**) Median scaled intensities of the 'type' markers across T cell subpopulations in the septum. The horizontal gray bars show the percentage of each cluster out of the total investigated cells. The table on the right shows the tentative biological names of the T cell subpopulations. (**b**) Percentage of each annotated cell population out of the total number of T cells (CD4+ and CD8+ T cells) from the septum of controls, dementia and MS donors. (**c**) Median scaled intensities of the 'state' markers in γδ T cells across all septum samples. Column annotation shows disease groups; row annotation shows adjusted p-values of comparing MS vs dementia and MS vs control samples. Color key shows row Z score. Blank cells are shown in a sample where γδ T cells were absent. Boxplots show median expression in γδ T cells of markers showing differential expression between the MS and control samples. ** adjusted p<0.05. Con: controls; Dem: dementia; MS: multiple sclerosis. Source data is listed in *Source data 1*.

The online version of this article includes the following figure supplement(s) for figure 2:

**Figure supplement 1.** Phenotyping the T cell populations in the septum using mass cytometry.

**Figure supplement 2.** Correlations between the percentage of T cells and different clinical parameters.

was lower in both disease groups than in controls (MS vs. controls: unadjusted p=0.04; dementia vs. controls: unadjusted p=0.09). Interestingly, γδ T cells in MS patients displayed reduced expression of the transcription factors T-bet and GATA3 compared to the controls (*Figure 2c*), suggesting a less activated phenotype (*Yin et al., 2002*). Notably, we detected some moderate negative correlations between the percentage of T cell subsets and PMD, sample delay and age (*Figure 2—figure supplement 2*). In addition, we observed higher abundance in males for CD8+ CD103- T_RM cells (cluster CD8 c1) and transitional CD8+ T cells (cluster CD8 c5).

Thus, similarly to other brain areas (*Smolders et al., 2018*), the septum is mainly comprised of tissue-resident memory cells and, to a lesser extent, effector memory T cells. The proportions of T cells subsets were equal among disease groups, suggesting that the accumulation of T cells in the MS

septum does not result from the enrichment of a particular T cell subset we were able to detect but from an accumulation of all of them.

## Accumulation of activated CD56[bright] NK cells and antibody-secreting cells in the MS septum

The higher percentage of NK cells observed in MS led us to further investigate this cell population using markers for cell migration and activation. Phenotypically, NK cells from MS donors displayed a more migratory profile than those from control and dementia cases (*Figure 3a*): higher expression of CD49d (integrin alpha-4), CD54 (ICAM1 or intercellular adhesion molecule 1), and CD31 (PECAM1 or platelet endothelial cell adhesion molecule). Contrarily, the expression of CD45RA and T-bet, markers of immature and cytotoxic NK cells, respectively (*Braakman et al., 1991*; *Townsend et al., 2004*), was lower in NK cells from MS donors compared to both the control and dementia groups.

Next, we further analysed NK cells together with B cells, as they share the expression of markers such as CD38, T-bet and CD27, and divided the two populations into smaller subsets (*Figure 3—figure supplement 1a-b*). Two NK cell populations have been described in humans based on their expression of CD56 and CD16 (*Mimpen et al., 2020*). In the septum, we identified both CD56[dim] CD16[+] NK cells and CD56[bright] CD16[-] NK cells in similar frequencies, two subsets of B cells (CD19[+] CD27[-] CD38[-] CD11c[-] T-bet[-] and CD19[+] CD27[-] CD38[-] CD11c[+] T-bet[+]) and another of antibody-secreting cells (ASCs; CD19[+] CD27[+] CD38[+]) (*Figure 3b* and *Figure 3—figure supplement 1c*).

CD56[bright] NK cells in the septum were more abundant in MS compared to the other two groups (*Figure 3c*). Moreover, CD56[bright] NK cells in MS donors expressed higher levels of adhesion molecules (CD49d, CD54 and CD31), ligand PDL1, costimulatory molecule CD86, and lower levels of CD45RA and T-bet relative to control and dementia cases (*Figure 3d*). Further sub-clustering of CD56[bright] NK cells revealed the existence of a Tbet[+] and a Tbet[-] subset (*Figure 3—figure supplement 2a-b*). Tbet[-] NK cells expressed higher levels of molecules that were increased in the general CD56[bright] NK cluster of MS patients (*Figure 3a*), and were the dominant CD56[bright] cell type in MS but not in control or dementia cases (*Figure 3—figure supplement 2c*). Overall, this suggests that specifically Tbet[-] CD56[bright] NK cells are increased in the septum of MS donors.

We also observed changes in the B cell lineage of the MS septum (*Figure 3c*). The frequency of B cells was higher in MS relative to dementia, but not relative to controls. Of note, B cells in the dementia group displayed an activated phenotype with higher expression of CD25 and CD49d (*Figure 3—figure supplement 3*). ASCs were rare in both dementia and controls and were a unique feature of the MS septum (*Figure 3c*). In addition, some moderate correlations existed between the percentage of NK and B cell subsets and PMD, sample delay and age (*Figure 3—figure supplement 4*).

Together, our data suggest that in MS, CD56[bright] NK cells either infiltrate the brain more efficiently than their CD56[dim] counterpart and/or proliferate at the septum leading to CD56[bright] NK cell accumulation. Moreover, we show that ASCs are also a typical feature of the MS septum, as seen in other brain regions (*Machado-Santos et al., 2018*).

## Activated CD56[dim] NK cells at the choroid plexus in MS

The central location of the septum adjacent to the brain ventricles guarantees a high exposure to the CSF. In MS patients, the immune environment in the CSF is altered (*Cepok et al., 2001*; *Rodríguez-Martín et al., 2015*). Here, we wanted to investigate if immune cell alterations in the MS septum correlate with those in the main producer of CSF: the choroid plexus. Demographic and clinical information of the patients is summarised in *Supplementary file 1* and *Figure 4—figure supplement 1*.

First, we wanted to identify and characterize the main immune cell populations in the choroid plexus. Overall, 9 immune cell populations were identified (*Figure 4a–b*) after manual merging of the clusters detected by the FlowSOM algorithm (*Figure 4—figure supplement 2*), with myeloid APCs (macrophages and dendritic cells) being the most abundant, followed by granulocytes. We also detected CD8[+] and CD4[+] T cells, NK cells, monocytes and cells from the B cell lineage. Unlike at the septum, the frequencies of the main immune cell subsets at the choroid plexus were similar among the disease groups (*Figure 4c*).

We found the choroid plexus to be populated with a variety of T cells, the most abundant subset being CD8[+] tissue-resident memory T cells (CD8 c1) (*Figure 5a* and *Figure 5—figure supplement*

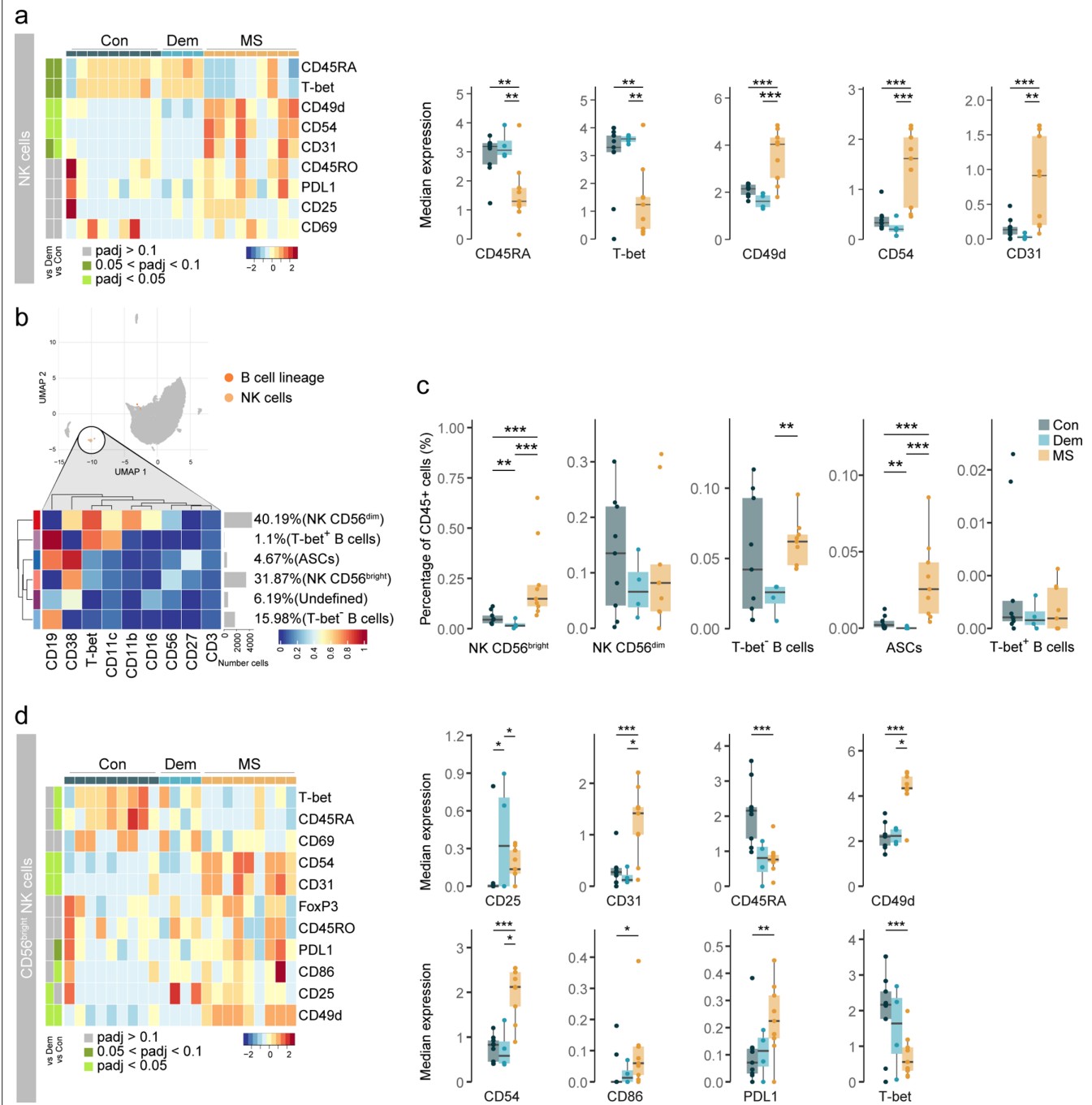

**Figure 3.** Activated CD56^bright NK cells and antibody-secreting cells accumulate in the MS septum. (**a**) Median scaled intensities of the 'state' markers in NK cells across all septum samples. Column annotation shows disease groups; row annotation shows adjusted p-values of comparing MS vs dementia and MS vs control. Color key shows row Z score. Boxplots show median expression in NK cells of markers with differential expression between MS and control samples. (**b**) Median scaled intensities of the 'type' markers across NK cell and B cell lineage populations in the septum. The horizontal gray bars show the percentage of each cluster out of the total investigated cells. (**c**) Percentage of each annotated cell population out of the total CD45^+ cells from the septum of controls, dementia, and MS donors. (**d**) Median scaled intensities of the 'state' (phenotype) markers in CD56^bright NK cells across all septum samples. Column annotation shows disease groups; row annotation shows adjusted p-values of comparing MS vs dementia and MS vs control samples. Color key shows row Z score. Boxplots on the right show median expression in CD56^bright NK cells of markers showing differential expression between the MS and control samples. NK: natural killer; ASCs: antibody-secreting cells. * adjusted p<0.1; ** adjusted p<0.05, *** adjusted p<0.01. Con: controls; Dem: dementia; MS: multiple sclerosis. Source data is listed in *Source data 1*.

The online version of this article includes the following figure supplement(s) for figure 3:

*Figure 3 continued on next page*

*Figure 3 continued*

**Figure supplement 1.** Subclustering of NK and B cell populations in the septum.

**Figure supplement 2.** NK cell subclustering in the septum.

**Figure supplement 3.** Marker expression in T-bet⁻ B cells from the septum.

**Figure supplement 4.** Correlation plots between the percentage of NK and B cell subsets and clinical parameters.

*1a*). Unlike the subset of CD8⁺ CD103⁺ T_RM cells from the septum (cluster CD8 c2), those in the choroid plexus did not express CD103 (cluster CD8 c1). We also identified double-negative T cells, and rare CD4⁺ T_REGS, suggesting a low abundance of these cells in the choroid plexus. As seen in the septum, all disease groups showed a similar frequency of the T cell subsets (*Figure 5—figure supplement 1b*).

NK cells from the MS choroid plexus expressed higher levels of the IL-2 receptor α-chain CD25 and of adhesion molecule CD54 relative to those from controls and dementia donors (*Figure 5b*). A closer look at the NK and B cell lineages (*Figure 5c* and *Figure 5—figure supplement 1c*) revealed that CD56^dim NK cells were more abundant than CD56^bright NK cells in the choroid plexus in all groups. Unlike in the septum, ASCs were rare in the choroid plexus regardless of disease. Although the frequencies of NK or B cell lineage cells were similar among the groups, there was a trend to a higher percentage of CD56^bright NK cells in MS donors relative to both the control and dementia groups (*Figure 5d*). However, it was the CD56^dim subset that expressed higher levels of CD54 in MS than in controls and dementia cases (*Figure 5e*).

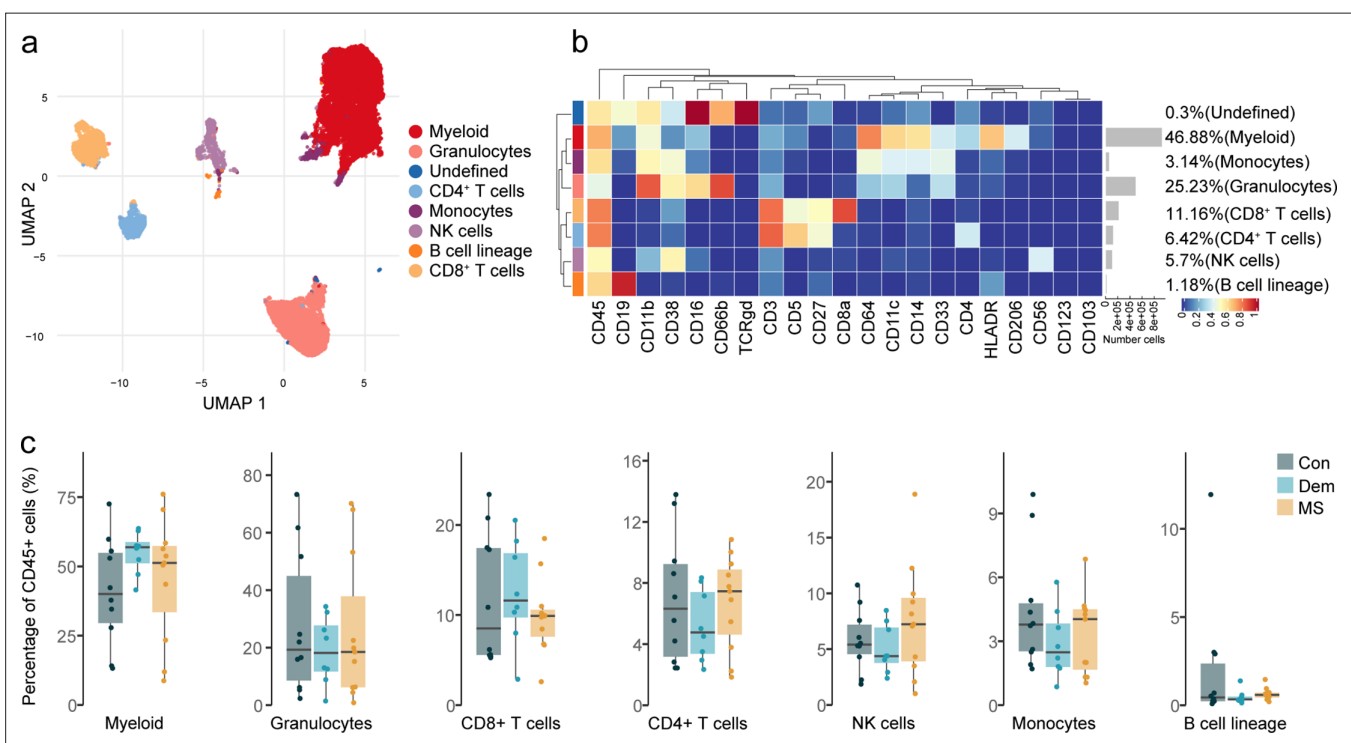

**Figure 4.** Immune phenotyping of the choroid plexus using mass cytometry. (**a**) UMAP plot based on the arcsinh-transformed expression of the 'type' (lineage) markers in the choroid plexus-derived immune cells. A subset of 1000 randomly selected cells per sample is shown, colored according to the manually annotated clusters. (**b**) Median scaled intensities of the 'type' (lineage) markers across choroid plexus-derived immune cell populations. The horizontal gray bars show the percentage of each cluster out of the total number of cells. (**c**) Percentage of each annotated cell population out of the total number of CD45⁺ cells from the choroid plexus of controls, dementia and MS donors. Con: controls; Dem: dementia; MS: multiple sclerosis. Source data is listed in *Source data 2*.

The online version of this article includes the following figure supplement(s) for figure 4:

**Figure supplement 1.** Demographic and clinical information.

**Figure supplement 2.** Immune phenotyping of the choroid plexus using mass cytometry.

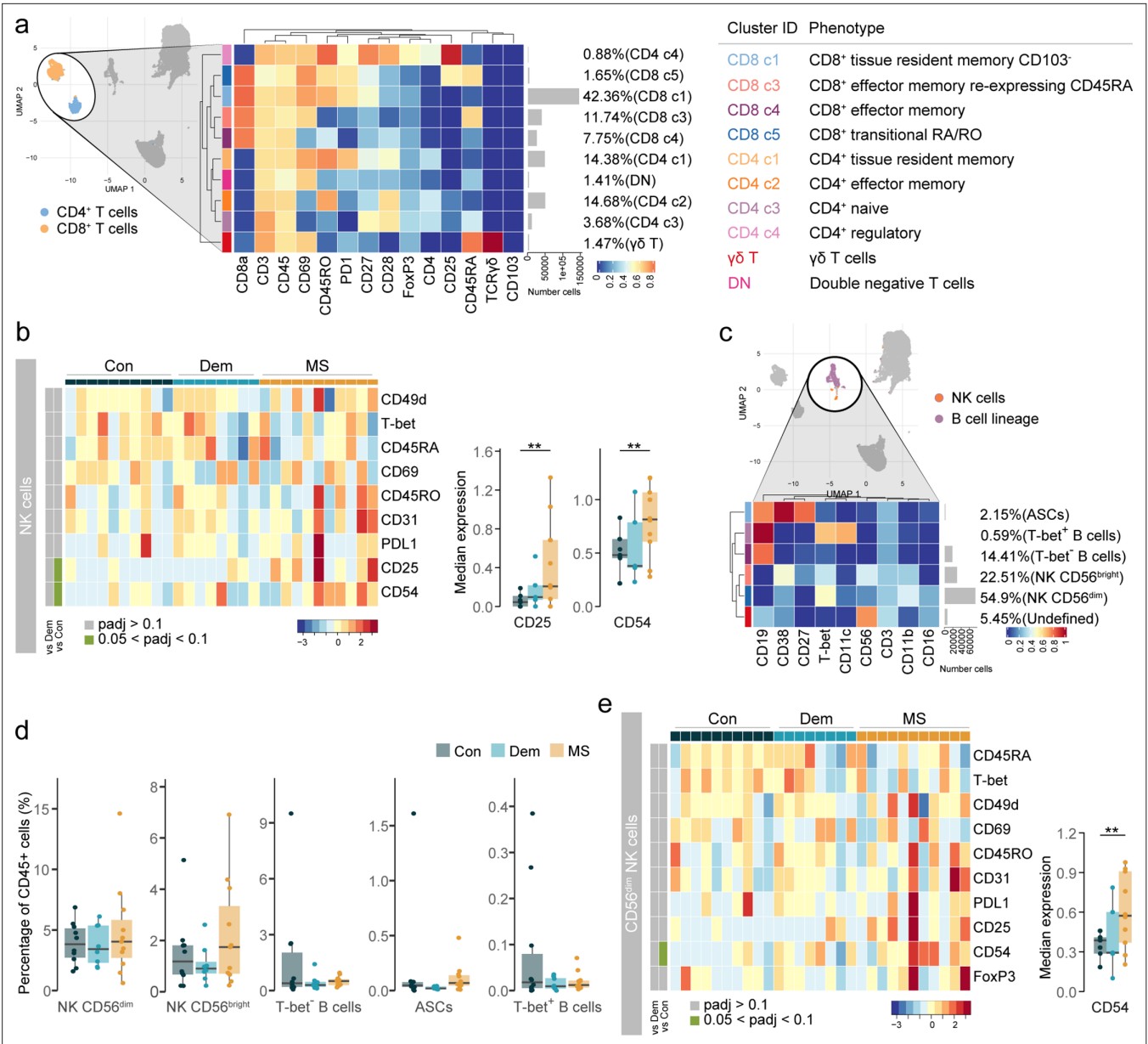

**Figure 5.** The choroid plexus in MS presents alterations in the NK cell populations. (**a**) Median scaled intensities of the 'type' markers across T cell subsets in the choroid plexus. The table shows tentative biological names of the T cell subpopulations. (**b**) Median scaled intensities of the 'state' markers in NK cells across all choroid plexus samples. Column annotation shows disease groups; row annotation shows adjusted p-values of comparing dementia vs controls and MS vs control samples. Color key shows row Z score. Boxplots show median expression in NK cells of markers with differential expression between MS and controls. (**c**) Median scaled intensities of the 'type' markers across NK cell and B cell lineage populations in the choroid plexus. (**d**) Percentage of each annotated NK or B cell population out of the total number of CD45+ cells from the choroid plexus of controls, dementia, and MS donors. (**e**) Median scaled intensities of the 'state' markers in CD56dim NK cells across all choroid plexus samples. Column annotation shows disease groups; row annotation shows adjusted p-values of comparing dementia vs control and MS vs control samples. Color key shows row Z score. Boxplot shows median expression in CD56dim NK cells of the marker with differential expression between MS and control. ** adjusted p<0.05. Con: controls; Dem: dementia; MS: multiple sclerosis. (**a, c**) The horizontal gray bars show the percentage out of the total cells. Source data is listed in *Source data 2*.

The online version of this article includes the following figure supplement(s) for figure 5:

**Figure supplement 1.** Immune phenotyping of the T cell, NK and B cell subsets in choroid plexus.

In sum, myeloid APCs are the most abundant immune cell population in the choroid plexus. Our high-dimensional approach allowed us to better define the subsets of T cells and other less abundant immune cells such as NK cells. Interestingly, at the MS choroid plexus, we found a phenotype shift in the CD56[dim] NK cells which did not result in a different abundance in the tissue. While there only was a non-significant trend for more CD56[bright] NK cells, we cannot discard the possibility that the choroid plexus contributes to the infiltration of CD56[bright] NK cells in the MS septum.

## Circulating CD56[bright] NK cells show a migratory and activated phenotype in MS

The high percentage of granulocytes and monocytes detected in the choroid plexus suggested a strong contribution from peripheral cells circulating in its abundant vascular network. To study this, we also performed CyTOF on blood-derived immune cells from the same donors. Demographic and clinical information of the patients is summarised in *Supplementary file 1* and *Figure 6—figure supplement 1* and the correspondent clusters detected by the FlowSOM algorithm after manual merging can be seen in *Figure 6—figure supplement 2a-c*. Indeed, when immune cells from all tissues were plotted in a UMAP, high similarities between choroid plexus and blood-derived immune cell populations were revealed (*Figure 6a* and *Figure 6—figure supplement 2d*). The septum, on the contrary, contained mostly brain-resident microglia and other tissue-resident immune cells.

In blood, granulocytes comprised the main cell type (*Figure 6b–d , and* ). NK cells were the second most abundant population, followed by T cells and monocytes. Of note, T cell subsets were present at a CD4:CD8 ratio of 0.76:1, instead of the expected 2:1, possibly due to the postmortem delay (*Ferreira et al., 2018*). Circulating B cells were more frequent in MS blood compared to control and dementia donors (*Figure 6d*). The frequency of NK cells in the choroid plexus positively correlated with that in the septum and blood (*Figure 6f* and *Figure 6—figure supplement 2e*), opening up the possibility that NK cells might traffic through the blood-CSF barrier.

We detected different populations of circulating T cells, including some that were absent in tissues such as CD8[dim] T cells (cluster CD8 c6), central memory CD4[+] T cells (T_{CM}, cluster CD4 c5) and CD4[+] T_{EMRA} (cluster CD4 c6) (*Figure 7a* and *Figure 7—figure supplement 1a-b*). The shift in B cells in MS was mostly due to T-bet[-] B cells (*Figure 7b–c*), but the same trend was observed in T-bet[+] B cells (MS vs. controls: unadjusted p=0.06; MS vs. non-neurological controls: unadjusted p=0.09). Interestingly, T-bet[+] B cells expressed higher levels of T-bet and CD31 in MS relative to controls (*Figure 7d*).

Most NK cells in the blood were CD56[dim] (*Figure 7b–c* and *Figure 7—figure supplement 2a*), and their percentage strongly correlated with that of the choroid plexus (*Figure 7—figure supplement 2b*). The abundance of NK cells was similar among disease groups, but the CD56[bright] NK cluster from MS blood expressed higher levels of the adhesion molecules CD54 and CD31 (*Figure 7e*), in line with the phenotype of CD56[bright] NK cells in MS septum. This could indicate preferential migration of the CD56[bright] subset from blood to the septum.

Correlations of immune cell abundance between the different tissues revealed a different pattern in controls than in MS (*Figure 7f*). In MS donors, there were more connections between immune cells derived from the septum, choroid plexus and the blood, suggesting increased cellular communication and movement in periventricular areas in MS. For example, naïve CD4 T cells (CD4 c3) from blood are negatively associated with naïve CD4 T cells in the MS septum but not in non-neurological controls, suggesting some form of trade-off mechanism or migration that is not seen in controls.

Overall, despite the high vascularization of the choroid plexus, we observed a different immune profile in the blood compared to that in the choroid plexus. First, the higher frequency of circulating B cells in MS relative to controls was not observed in the choroid plexus; second, the migratory phenotype of CD56[dim] NK cells was altered in MS choroid plexus but not in MS blood; third, we show that circulating CD56[bright] NK cells but not their CD56[dim] counterpart, display a migratory phenotype in MS. In summary, we identified several associations among the immune populations of the septum, the choroid plexus and the blood, indicative of a dynamic immune environment in periventricular tissues.

## Granzyme K[+] NK cells accumulate in periventricular lesions and choroid plexus from MS donors

Finally, to confirm that CD56[bright] NK cells accumulate in periventricular brain regions in MS donors, we used multiplex immunohistochemistry in an independent cohort (*Supplementary file 1*), wherein

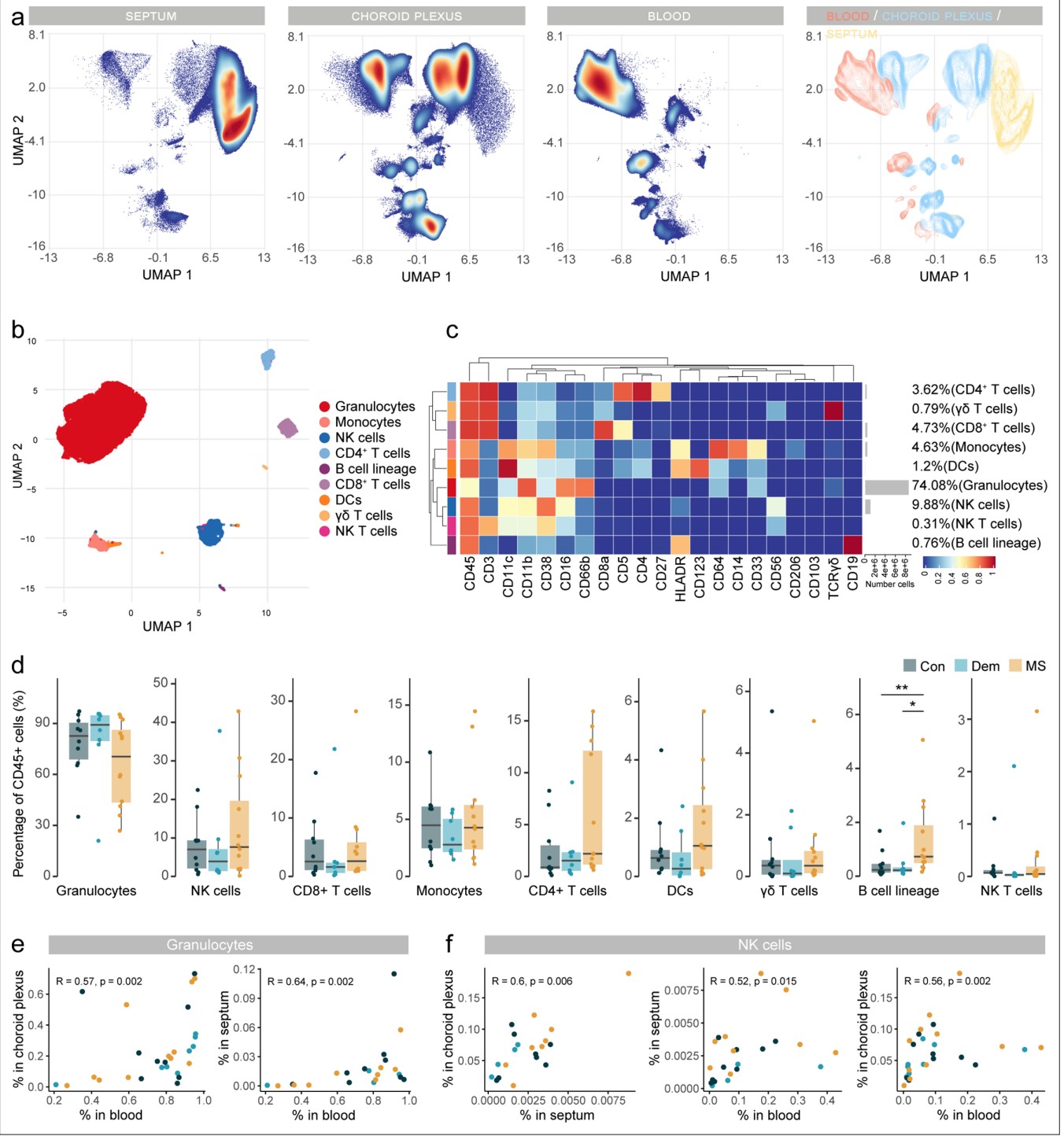

**Figure 6.** Immune phenotyping reveals an accumulation of B cells in postmortem MS blood. (**a**) UMAP plot based on the arcsinh-transformed expression of all markers in the immune cells of the septum, choroid plexus, and blood. A total of 100,000 randomly selected cells were selected per sample. The first three plots are colored according to the density. The final plot shows an overlay figure where cells are colored by their tissue of origin. (**b**) UMAP plot based on the arcsinh-transformed expression of the 'type' (lineage) markers in the blood-derived immune cells. A subset of 1000 randomly selected cells per sample is shown, colored according to the manually annotated clusters. (**c**) Median scaled intensities of the 'type' markers across blood-derived immune cell populations. Horizontal gray bars show the percentage out of the total cells. (**d**) Percentage of each annotated cell population out of the total number of CD45+ cells from the blood of control, dementia, and MS donors. (**e**) Scatter plots showing correlations between the proportions of granulocytes in blood, choroid plexus, and septum. Each dot represents a sample, colored by disease group. (**f**) Scatter plots showing correlations between the proportions of NK cells in different tissues. Each dot represents a sample, colored by disease group. R: Spearman's

*Figure 6 continued on next page*

*Figure 6 continued*

Rho rank correlation coefficient; p=p-value. * adjusted p<0.1; ** adjusted p<0.05. Con: controls; Dem: dementia; MS: multiple sclerosis. Source data for a, e, f is listed in *Source data 1–3*; for b, c, d in *Source data 3*.

The online version of this article includes the following figure supplement(s) for figure 6:

**Figure supplement 1.** Demographic and clinical information.

**Figure supplement 2.** Immune phenotyping of the main immune cell populations in blood and their correlations with those in septum and choroid plexus.

MS and control groups were age-matched (Wilcoxon rank-sum test with continuity correction, p-value = .41) and had a similar female:male ratio (0.667 in controls and 0.75 in MS). This technique allowed us to stain up to seven markers in the same section. Due to the difficulty of making a clear distinction between bright and dim expression of CD56 in NK cells by immunohistochemistry, we used granzymes (Gr) expression as a surrogate marker. We defined CD56$^{bright}$ NK cells as CD45$^+$ NKp46$^+$ GrK$^+$ GrB$^-$ and CD56$^{dim}$ NK cells as CD45$^+$ NKp46$^+$ GrK$^-$ GrB$^+$ (*Bratke et al., 2005*; *Hanna et al., 2004*).

We analysed the presence of NK cells in periventricular brain tissue from controls and within normal-appearing white matter (NAWM) and white matter lesions from MS donors. Lesions were identified by the absence of myelin proteolipid protein and the abundance of HLA-DR$^+$ cells (*Figure 8a*). GrK$^+$ NK cells accumulated in both the border and the centre of the lesion compared to NAWM and were more abundant than GrB$^+$ NK cells (*Figure 8b–c*). The increased presence of GrK$^+$ NK cells in the MS lesion confirms our earlier CyTOF data where the MS septum contained more CD56$^{bright}$ than CD56$^{dim}$ NK cells, while CD56$^{dim}$ NK cells were the dominant NK subset in septum derived from controls and donors with dementia.

In addition, the use of choroid plexus tissue sections allowed us to specifically look at infiltrated cells and discard those in the circulation that confounded our CyTOF analysis. In this cohort, we observed a significantly higher density of GrK$^+$ but not GrB$^+$ NK cells in the choroid plexus of MS vs. control donors (*Figure 8b and d*), indicating that the increase shown using CyTOF was specific to the tissue and not due to the high vascularization of the choroid plexus.

In sum, we validated the accumulation of NK cells within the periventricular brain of MS donors. In particular, GrK$^+$ NK cells were more abundant and enriched in MS lesions and the choroid plexus stroma.

## Discussion

In this study, we used a CyTOF multi-parameter approach to provide a comprehensive overview of the immune landscape in human periventricular brain tissue from donors with MS, dementia and controls.

Here, we revealed the involvement of CD56$^{bright}$ NK cells in MS periventricular pathology. Since periventricular areas of the brain are a predilection site for MS lesions (*Brownell and Hughes, 1962*; *Jehna et al., 2015*; *Palmer et al., 1999*), we focused on the septum as it separates the two lateral ventricles and is highly exposed to CSF. Although magnetic resonance imaging studies demonstrated the presence of MS lesions in the septum (*Gean-Marton et al., 1991*; *Palmer et al., 1999*; *Palotai et al., 2018*; *Simon et al., 1986*), the composition of the inflammatory infiltrates remained unknown. We found an expansion of CD56$^{bright}$ NK cells within the septum of MS donors, and these CD56$^{bright}$ NK cells expressed higher levels of proteins involved in NK cell activation and migration. Using multiplex immunohistochemistry, we validated the accumulation of NK cells expressing GrK, typical of the CD56$^{bright}$ subset (*Bratke et al., 2005*), in periventricular MS lesions from an independent cohort. Putative CD56$^{bright}$ NK cells abound in the rim of the lesions, where the highest activity is present in terms of immune cell activation and demyelination. The centre of MS lesions is characteristically hypocellular, and this is reflected in the number of NK cells present. We also found a slight increase in the density of GrB$^+$ NK cells — the putative CD56$^{dim}$ cells — in MS lesions versus NAWM and healthy controls that was not detected with the CyTOF. Next to NK cells, we detected higher frequencies of ASCs, CD4$^+$ and CD8$^+$ T cells in the septum from MS donors compared to controls, which is similar to other periventricular regions (*Machado-Santos et al., 2018*). The distribution of general immune cell populations in the septum was consistent with previous literature (*Korin et al., 2017*; *Smolders et al., 2018*).

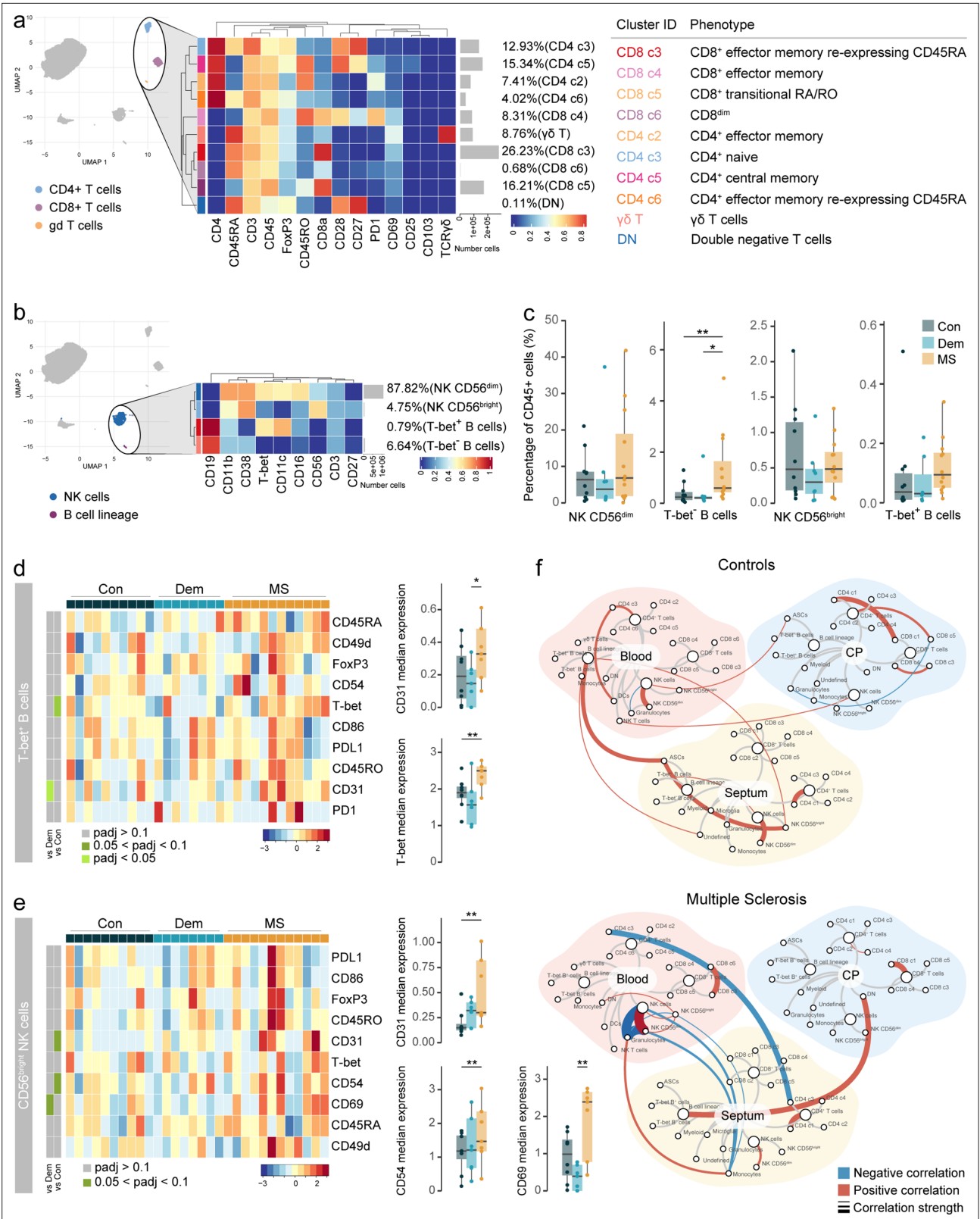

**Figure 7.** Circulating Tbet⁺ B cells in MS express higher levels of T-bet. (**a**) Median scaled intensities of the 'type' markers across T cell subpopulations in blood. Table shows tentative biological names of the T cell subpopulations. (**b**) Median scaled intensities of the 'type' markers across NK cell and B cell lineage populations in blood. (**c**) Percentage of NK and B cell lineage populations out of the total CD45⁺ cells from the blood of control, dementia and MS donors. (**d**) Median scaled intensities of the 'state' markers in T-bet⁺ B cells across all blood samples. Column annotation shows disease groups;

*Figure 7 continued on next page*

Figure 7 continued

row annotation shows adjusted p-values of comparing dementia vs control and MS vs control. Color key shows row Z score. Boxplots show median expression in T-bet⁺ B cells of markers with differential expression between MS and control. (**e**) Median scaled intensities of 'state' markers in CD56^bright NK cells across all blood samples. Column annotation shows disease groups; row annotation shows adjusted p-values of comparing dementia vs control and MS vs control samples. Color key shows row Z score. Boxplots show median expression in CD56^bright NK cells of markers with differential expression between MS and control. (**f**) Network plot illustrating correlations between identified immune subsets in the three anatomical sites analysed. Positive correlations are displayed in red and negative correlations in blue. Width of the line is proportional to the correlation coefficients. p-Values adjusted with Benjamini-Hochberg method. ** adjusted p<0.05. Con: controls; Dem: dementia; MS: multiple sclerosis (**a, b**) Horizontal gray bars show the percentage out of the total cells. Source data for a, b, c, d, e is listed in *Source data 3*; for f in *Source data 1–3*.

The online version of this article includes the following figure supplement(s) for figure 7:

**Figure supplement 1.** Immune phenotyping of the T cell, NK and B cell subsets in blood and their correlations with those in septum and choroid plexus.

**Figure supplement 2.** Median intensity of NK and B cell subsets in blood and correlations of their proportions with those in septum and choroid plexus.

NK cells are lymphocytes of the innate immune system with the ability to regulate adaptive responses. There are two main NK cell subsets based on the expression of CD56 and CD16: CD56^bright NK cells express low levels of CD16 and are specialised immunoregulators, while CD56^dim NK cells have a potent cytolytic capacity, which may be mediated by CD16 (*Mimpen et al., 2020*). Importantly, we relied on the markers CD56 and NKp46 to identify human NK cells by CyTOF and immunohistochemistry, respectively. However, NK cells belong to the broader group of innate lymphoid cells (ILCs), which also include ILC1, ILC2 and ILC3 subsets. ILC subpopulations are highly plastic and their identification remains a challenge; it has been proposed that they belong to a single, highly plastic population, but this debate is beyond the scope of this study. Although NK cells are thought to predominate in the brain parenchyma (*Sedgwick et al., 2020*), we cannot exclude the presence of other ILCs. Particularly, certain subsets of ILC3s share the expression of CD56 with NK cells (*Trabanelli et al., 2018*). The extent and pathological consequences of the presence of ILC3s in MS should be determined in future studies.

A beneficial role of CD56^bright NK cells in MS was first discovered with the anti-inflammatory drug daclizumab. Daclizumab is an anti-CD25 antibody therapy that leads to the expansion of circulating CD56^bright NK cells (*Bielekova et al., 2006*; *Wynn et al., 2010*), an increase that was later reported with other disease-modifying treatments for MS as well (*Gross et al., 2016b*; *Saraste et al., 2007*; *Smith et al., 2018*). In line with this, a protective role for NK cells was shown in animal models of MS (*Hao et al., 2010*; *Xu et al., 2005*) through cytotoxicity against autoreactive T cells (*Xu et al., 2005*). Despite the beneficial immunomodulatory role assigned to CD56^bright NK cells in the context of MS, the presence of NK cells in the CNS can also be detrimental (*Jin et al., 2021*; *Lagumersindez-Denis et al., 2017*; *Liu et al., 2016*; *Liu and Shi, 2019*; *Morse et al., 2001*). Thus, while NK cells may act as a double-edged sword in MS (*Gross et al., 2016a*), we postulate that the presence of specifically CD56^bright NK cells in the periventricular areas primarily has a beneficial effect on MS pathology by limiting neuroinflammation.

To date, studies of NK cells in the MS brain parenchyma did not distinguish between the CD56^bright and CD56^dim NK subsets. NK cells were detected in the brains from MS donors (*Gross et al., 2016a*; *Lagumersindez-Denis et al., 2017*; *Liu et al., 2016*; *Traugott and Raine, 1984*), while absent from controls (*Gross et al., 2016a*; *Liu et al., 2016*). Within active MS lesions, NK cells express GrK polarised towards neighbouring T cells (*Gross et al., 2016a*; *Jiang et al., 2011*). Thus, the expansion of the CD56^bright NK cell population we observed in MS brains may mediate a protective response to neuroinflammation by killing autoreactive T cells through the release of GrK. Moreover, we observed higher expression of the ligand PDL1 in CD56^bright NK cells from MS brains compared to those from controls, suggesting an involvement of the PD1-PDL1 axis. PDL1 is an NK cell activation marker associated with enhanced cytotoxicity (*Dong et al., 2019*) and binds the T cell co-inhibitory molecule PD1 inducing suppression of PD1⁺ autoreactive T cells (*Riley, 2009*).

Proportions of circulating CD56^bright NK cells remained the same in MS and control donors, as described before (*Gross et al., 2016a*; *Laroni et al., 2016*). Interestingly, CD56^bright NK cells in both the MS brain and blood expressed higher levels of proteins associated with cell migration. As such, the enrichment of CD56^bright NK cells in the CNS of MS patients could result from selective infiltration from

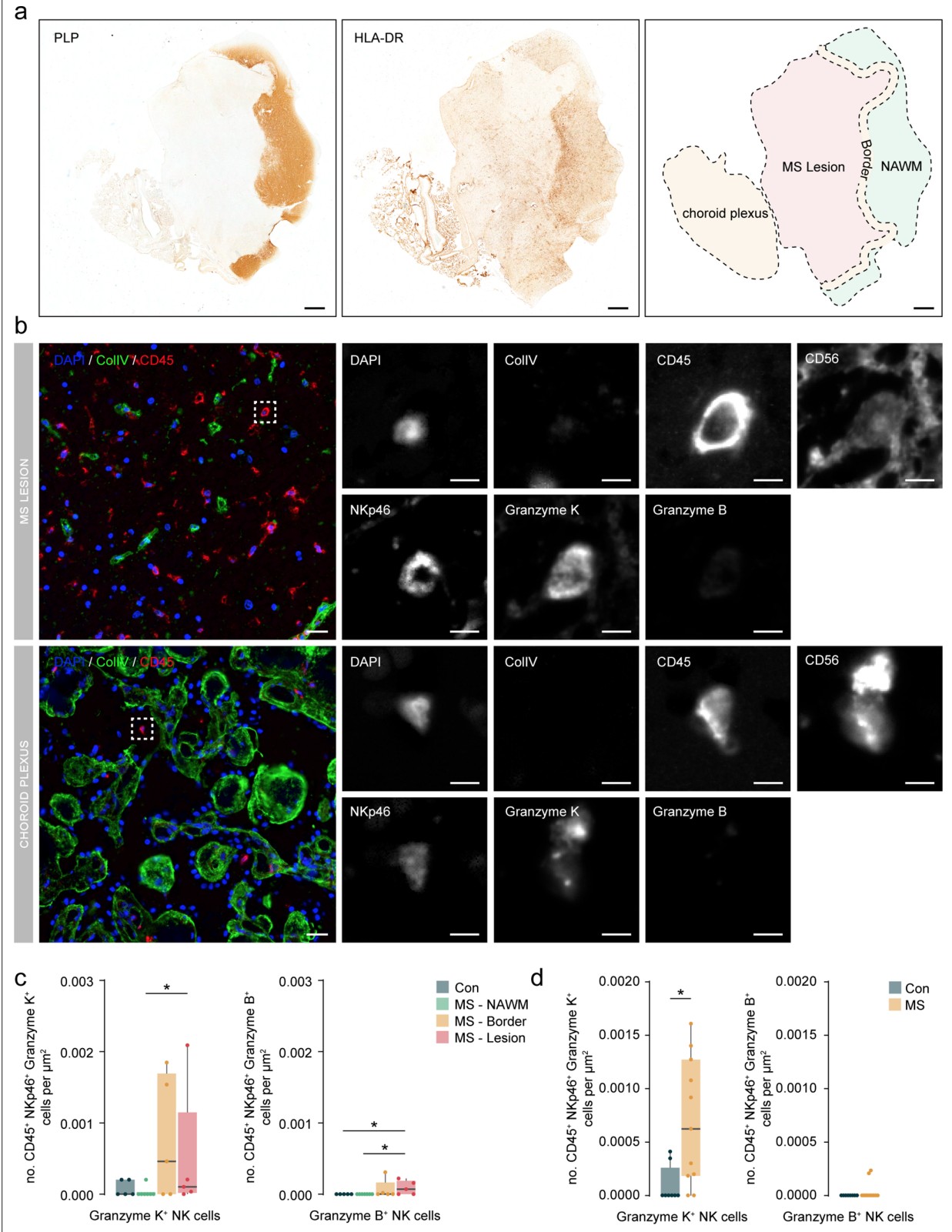

**Figure 8.** Granzyme K[+] NK cells accumulate in periventricular lesions and choroid plexus. (**a**) Representative immunohistochemical staining of myelin with proteolipid protein (PLP) and presence of immune cells with HLA-DR. The overview shows the location of the normal-appearing white matter (NAWM), the lesion border, the lesion, and the choroid plexus. Scale bar = 1 mm. (**b**) Representative multiplex immunohistochemical staining of MS lesion and the choroid plexus. The merged overview shows DAPI (blue), ColIV (green), and CD45 (red). The dotted box indicates CD45[+] NKp46[+]

*Figure 8 continued on next page*

*Figure 8 continued*

Granzyme K$^+$ cell. A magnified image of all separate channels is shown in grayscale. Scalebar = 25 um for overview and 5 um for magnifications. (**c**) Boxplots display the abundance of CD45$^+$ NKp46$^+$ Granzyme K$^+$ and CD45$^+$ NKp46$^+$ Granzyme B$^+$ cells per µm$^2$ in MS lesions. Con: controls; MS: multiple sclerosis; NAWM: normal-appearing white matter; * p<0.05. (**d**) Boxplots display the abundance of CD45$^+$ NKp46$^+$ Granzyme K$^+$ and CD45$^+$ NKp46$^+$ Granzyme B$^+$ cells per µm$^2$ in choroid plexus. Con: controls; MS: multiple sclerosis. * p<0.05.

the blood towards the brain rather than being the result of accumulation in the circulation. In line with this, migration studies showed that CD56$^{bright}$ NK cells have a higher capacity to transmigrate across a blood-brain barrier in vitro model than their CD56$^{dim}$ counterparts (*Gross et al., 2016a*). However, this study did not show enhanced migration of CD56$^{bright}$ NK cells from MS donors compared to controls.

Direct infiltration from the circulation through the local brain vasculature is a likely route for MS immune infiltrates. Besides, abundant periventricular veins drain to the ventricles (*Pardini et al., 2016*), resulting in an indirect peripheral-brain connection through the choroid plexus via the CSF. Here, we described the presence of NK cells in the human choroid plexus and identified the upregulation of the adhesion molecule CD54 in CD56$^{dim}$ NK cells in MS. The higher ratio of CD56$^{bright}$/CD56$^{dim}$ cells in the choroid plexus compared to the blood indicates enrichment of CD56$^{bright}$ NK cells in the choroid plexus. Accordingly, we could not detect GrB$^+$ NK cells, presumptive CD56$^{dim}$ NK cells, within the choroid plexus stroma from controls, and rarely in MS cases, using multiplex immunohistochemistry. The upregulation of CD54 in CD56$^{dim}$ NK cells in MS was unique to the choroid plexus and differed from CD56$^{dim}$ NK cells in the circulation and the septum. However, the proportion of CD56$^{dim}$ NK cells did not differ between groups. Although the higher frequency of CD56$^{bright}$ NK cells measured by CyTOF in the choroid plexus from MS donors relative to controls was not significant, multiplex immunohistochemistry confirmed the accumulation of GrK$^+$ NK cells in the choroid plexus stroma of MS donors from an independent cohort. Future studies should uncover if CD56$^{bright}$ NK cells continue their journey across the blood-CSF barrier and the ependymal layer lining the periventricular brain. Supporting this idea, the CSF ratio of CD56$^{bright}$/CD56$^{dim}$ NK cells is higher in MS patients than in controls (*Rodríguez-Martín et al., 2015*; *Schafflick et al., 2020*), and it is further increased upon treatment with the anti-CD25 antibody therapy daclizumab (*Bielekova et al., 2011*). In conclusion, circulating CD56$^{bright}$ NK cells in MS may infiltrate into periventricular brain areas directly across the blood-brain barrier, and indirectly via the choroid plexus-CSF route.

Our study has two main limitations, first scarcity of fresh human tissue prevented having sex and age-matched groups with large sample sizes for the CyTOF analysis. To overcome the underpowered design and possible effects of confounders, we have validated our main finding by multiplex immunohistochemistry in a separate cohort with a similar age and female/male ratio. Secondly, there is a strong contribution of blood-derived immune cells in the choroid plexus, which precluded a clear distinction between circulating and stromal immune cells. This may have prevented the detection of choroid-plexus specific changes in the stroma, such as an accumulation of CD8$^+$ T cells in the choroid plexus from MS donors, previously described by our group using immunohistochemistry (*Rodríguez-Lorenzo et al., 2020b*). In addition, the high proportion of granulocytes in the CP as detected by our CyTOF analysis likely originates from the circulation (*Rodríguez-Lorenzo et al., 2020b*; *Vercellino et al., 2008*). Contrariwise, the scarcity of B cells, despite the high vascularisation, is in line with previous reports (*Rodríguez-Lorenzo et al., 2020b*; *Vercellino et al., 2008*); and the detection of rare ASCs in the choroid plexus but not in the blood reassures their tissue specificity (*Vercellino et al., 2008*).

Notably, our study with postmortem blood uncovered a higher frequency of B cells in MS donors relative to controls and dementia cases. While the efficacy of peripheral B cell-depleting therapies suggests a pathogenic role of circulating B cells in MS (*Hauser et al., 2017*; *Montalban et al., 2017*), we could not find evidence in the literature showing an accumulation of B cells in the blood of MS patients, and a recent study showed similar frequencies relative to controls (*Schafflick et al., 2020*). Although most B cells lack T-bet expression, a small subset of T-bet$^+$ B cells shows a trend towards more abundance in MS postmortem blood, in line with recent literature (*Couloume et al., 2021*; *van Langelaar et al., 2019*). Moreover, the expression of T-bet within T-bet$^+$ B cells was higher in MS donors relative to control and dementia donors. T-bet expression on B cells has been associated with increased pathogenic responses (*Barnett et al., 2016*; *Piovesan et al., 2017*). The relevance of T-bet$^+$ B cells in MS should be further confirmed in fresh blood samples.

In addition, we aimed to uncover MS-specific immune changes by comparing our findings in MS donors to an extra neurological control group consisting of demented cases next to our non-neurological controls. All but one of our demented controls were diagnosed with Alzheimer's disease (AD). We found that the increase in CD56$^{bright}$ NK cells and ASCs were specific to MS compared to both dementia and controls. Although our study was not powered for focusing on dementia, we found that the abundance of CD8$^+$ T cells in the septal periventricular area was higher compared to the controls, following the trend seen in MS donors. The most notable increase was seen in CD8$^+$ tissue-resident memory T cells, a population that has been found in the ageing brain before (*Smolders et al., 2018*). In AD, higher numbers of circulating CD8$^+$ T$_{EMRA}$ cells relative to healthy donors were negatively associated with cognition (*Gate et al., 2020*). In addition, clonally expanded CD8$^+$ T cells were more abundant in the CSF of AD patients compared to healthy controls (*Gate et al., 2020*). Although our findings were not statistically significant, they highlight a role for periventricular CD8$^+$ T cells in AD pathology. Moreover, we identified an activated phenotype of periventricular B cells in demented donors, consisting of higher expression of CD25 and CD49d. Interestingly, an accumulation of activated B cells was found in the circulation of AD mouse models before, together with infiltration of B cells into the brain parenchyma (*Kim et al., 2021*). Future studies with increased sample size should confirm these results and reveal how T and B cells play a role in AD pathogenesis.

## Conclusions

In sum, our study provides a characterization of the periventricular immune landscape in MS and reveals the involvement of CD56$^{bright}$ NK cells in local MS brain pathology. Moreover, we explored the relative contributions of the choroid plexus and peripheral blood to the immune composition of the periventricular MS brain. Our findings highlight the importance of CD56$^{bright}$ NK cells in the CNS and their potential as a therapeutic target in MS. Instead of indiscriminately preventing immune cell surveillance in the CNS, an expansion of immunoregulatory CD56$^{bright}$ NK cells could selectively suppress neuroinflammation while minimizing side effects for patients suffering from MS.

# Materials and methods

## Human samples and study design

Fresh postmortem tissue was obtained from donors by rapid autopsy from the Netherlands Brain Bank. Donors included cases with clinically diagnosed and neuro-pathologically confirmed progressive MS (labeled MS, n=12), cases with dementia (labeled Dem, used as an additional control; n=8) and control cases without neurological or autoimmune diseases (labeled Con; n=10). Septum pellucidum and choroid plexus from the right lateral ventricle were stored in Hibernate-A medium (Thermo Fisher, #A1247501); and postmortem blood of all included cases was collected in EDTA-coated tubes (BD Biosciences, #367525). Postmortem delay was less than 24 hr and tissues were stored for less than 24 hr (sample delay) before starting the isolation of immune cells.

Formalin-fixed paraffin-embedded tissue from choroid plexus and periventricular areas was obtained from patients with clinically diagnosed MS (choroid plexus, n=10; periventricular areas, n=7) and non-neurological controls (choroid plexus, n=8; periventricular areas, n=5) by rapid autopsy from the Netherlands Brain Bank and Multiple Sclerosis Society Tissue Bank, funded by the Multiple Sclerosis Society of Great Britain and Northern Ireland, registered charity 207,495. All the MS periventricular area blocks contained lesions.

All donors or their next of kin provided fully informed consent for autopsy and use of material for research from Netherlands Brain Bank under ethical approval by the Medical Ethics Committee of the Free University Medical Center in Amsterdam (2009/148), project number 1,127. Relevant clinical information of the donors is summarised in *Supplementary file 1*.

## Sample processing

The choroid plexus was washed twice with ice-cold PBS and incubated with 1 mM EDTA pH 7.2 in HBSS (Thermo Fisher, #14175095) rotating for 1 hr at 37 °C to loosen the tight junctions of the epithelium, washed again and thoroughly cut with sharp scissors and digested with proteolytic enzymes (2 U/mL Liberase TL, Sigma Aldrich, #5401020001) and DNase (33 µg/mL; Sigma Aldrich, #11284932001) for 30 min at 37 °C, resuspending every 10 min. Digestion was stopped with RP-HE (RPMI 1640,

Thermo Fisher; 10% FCS (Corning, #35–079-CV)), 10 mM EDTA (ThermoFisher, #15575020), 20 mM HEPES (Gibco, #156300–056), 50 µm 2-mercaptoethanol (Gibco, #31350010) for 5 min on ice. The cell suspension was filtered through a 70 µm cell strainer (Life Sciences) to remove undigested pieces of tissue. Immune cells were isolated from this cell suspension using a 70–30% Percoll (Sigma Aldrich, #17-0891-01) gradient as follows. The eluent containing the single-cell suspension was centrifuged, the pellet was resuspended in 70% Percoll, and 30% Percoll was carefully layered on top. Following centrifugation at 900 g for 30 min at 22 °C, immune cells in the 70–30% interphase were collected and washed twice with RP10 (RPMI 1640, Thermo Fisher; 10% FCS, 1% penicillin/streptomycin (Invitrogen, #15140122), 1% glutamine (Thermo Fisher, #25030–024) and counted. For the septum, an identical isolation protocol was used leaving out the first step of incubation with EDTA.

Postmortem blood was diluted with 1% citrate in PBS and filtered through a 70 µm cell strainer to remove clots. Diluted and filtered blood was then carefully layered on top of Lymphoprep (STEMCELL Technologies, #07851). The gradient was centrifuged at 800 g for 30 min at room temperature. The immune cells in the interphase were collected and washed with 1% citrate in PBS. Erythrocytes were lysed with ACK Lysing Buffer (Thermo Fisher, #A1049201) for 5 min at room temperature. Lysis was stopped with 1% citrate in PBS and cells were washed twice with RP10 and counted.

## Viability staining, fixation, and freezing of immune cells

Isolated immune cells were washed with HBSS$^{-/-}$ (without Mg$^{+2}$, Ca$^{+2}$, and phenol red, #14175–095). Cells were distributed in wells of a 96-well V-bottom plate (0.5–1 million cells/well for septum and choroid plexus cells, 5 million cells/well for blood cells) and washed again with HBSS$^{-/-}$. Cells were washed with Maxpar PBS (Fluidigm, #201058) and stained with the viability marker Cell-ID Cisplatin-198Pt (Fluidigm, #201198) for 5 min at 37 °C according to the manufacturer's instructions. Cells were then washed three times with RP10 and fixed with Maxpar Fix I Buffer (Fluidigm, #201065) for 10 min at room temperature. Fixed cells were centrifuged at 800 g for 7 min at 4 °C. Cell pellets were resuspended in 10% DMSO in FCS and put in a Mr. Frosty Freezing Container at –80 °C for 24 hr. Samples were stored at –80 °C until staining and sample acquisition.

## Antibody labelling and titration

Antibody labeling with the indicated metal tag was performed using the MaxPAR antibody conjugation kit (Fluidigm) according to the manufacturer's instructions. Purification of the bound antibody was performed with high-performance liquid chromatography (Thermo Fisher) and subsequently concentrated by filtering with a 10 kDa filter (Merck Millipore) in a swing-out bucket at 4000 RPM for 15 min. The end volume was determined and an equal volume of antibody stabilizer buffer (Fluidigm; supplemented with 0.05% sodium azide) was added before the antibodies were stored at 4 °C. All antibodies used in this study were titrated using both fixed and unfixed thawed PBMCs and the most optimal concentrations with the least spillover were chosen. Concentrated antibody cocktails for surface and nuclear antigen detection were made, aliquoted, and stored at –80 °C, as previously described (*Schulz et al., 2019*).

## Staining protocol

During the staining, reagents were cooled on ice, centrifugation steps were performed at 800 g for 7 min at 4 °C with acceleration 9 and deceleration 7, and incubations were performed at room temperature. Samples were thawed rapidly and washed twice with 8 mL of Maxpar PBS (Fluidigm, #201058). Cell pellets were resuspended in Maxpar PBS and transferred to a 96 well V-bottom plate (Sigma Aldrich, #M9686). After centrifugation, cells were washed twice with 150 µL of 1 X Barcode Perm Buffer (Fluidigm, #201057). Then, samples were incubated with the appropriate barcodes (Fluidigm, #201060) in 1 X Barcode Perm Buffer for 30 min, gently mixing after 15 min. After centrifugation, samples were washed twice with 150 µL of cell staining buffer (CSB) (Fluidigm, #201068) and the cells from all samples were pooled. Cells in this combined sample were counted and after centrifugation, cells were incubated with FC block (BioLegend, #422302) diluted in CSB (1:50) for 10 min. Then, the surface antibody cocktail (thawed and centrifuged for 15 min at 15,000 x g at 4 °C) was diluted with CSB in the appropriate concentration and volume relative to the cell count (*Supplementary file 2*) and added to the combined sample. This was followed by a 30 min incubation, gently mixing after 15 min. After the incubation, cells were washed twice with CSB followed by a wash with Maxpar PBS.

Then, cells were fixed with 1 mL of freshly made 1.6% PFA (Thermo Fisher, #28906) in Maxpar PBS for 10 min. After centrifugation, cells were permeabilised with a 30 min incubation in 1 mL of FoxP3 Fix/Perm working solution (eBioscience, #00–5523), gently mixing after 15 min. After centrifugation, cells were washed twice with 1 X Permeabilisation Buffer (eBioscience, #00–5523). The nuclear antibody cocktail was thawed and centrifuged for 15 min at 15,000 x g at 4 °C and cells were incubated with the nuclear antibody cocktail in 1 X Permeabilisation Buffer with the appropriate concentration to the cell count. The cells were incubated with the nuclear antibody mix for 45 min, gently mixing every 15 min. Cells were then washed three times with 1 X Permeabilisation Buffer and fixed with 1 mL of freshly made 1.6% PFA in Maxpar PBS for 10 min. After centrifugation, nucleated cells were stained with Maxpar Intercalator (Fluidigm, #201,192B) diluted 1:4000 in Maxpar Fix and Perm Buffer (Fluidigm, #201067) overnight at 4 °C, until sample acquisition.

## Sample acquisition

Cells in the Intercalator solution were washed twice with CSB and divided over approximately 1 × $10^6$ cells per tube, followed by washing with cell acquisition solution (CAS) (Fluidigm, #201241) right before acquisition. Samples were filtered and calibration beads (Fluidigm, #201078) were added to the suspension to 15% of the final volume. Cells were acquired on the Helios (Fluidigm), with an event rate of 250–350 events per second. Runs took approximately 30–45 min. During the day, tuning of the machine was performed during start-up and after 4 hr of sample acquisition. Within each barcoded set of samples, one reference sample was included to monitor possible differences in staining intensity between barcodes due to technical variation in the staining protocol or daily changes in instrument functioning.

## Generation of the reference sample

The reference sample contained peripheral blood mononuclear cells (PBMCs) obtained from the blood of 3 healthy controls of which a part was stimulated with a cytokine cocktail to induce expression of each protein and transcription factor included in the CyTOF panel. Unstimulated and stimulated PBMCs were combined, stained for viability as described earlier, fixated, and stored in aliquots at –80 °C until further use.

## CyTOF data pre-processing and tissue comparisons

Acquired samples were randomised using Gaussian negative half zero randomization in CyTOF Software version 6.7. The FCS files were normalised using bead normalisation, concatenated, and debarcoded using the CyTOF Software version 6.7. The concatenated FCS files were uploaded into FlowJo V10 to perform the clean-up where normalization beads, cell debris, and cell doublets were removed using DNA, beads and Gaussian parameters. Next, live cells showing negative reactivity for viability marker Cell-ID Cisplatin-198Pt and dim-to-positive reactivity for CD45-89Y were selected and used for the further processing steps. To anticipate similarities between choroid plexus, septum, and blood-derived immune cells, 10,000 cells per donor tissue were imputed and visualised in one UMAP (Uniform Manifold Approximation and Projection).

## CyTOF data processing and statistical analysis

Data analysis was performed in R version 4.0.3 (2020-10-10). Clinical data were analysed by the Kruskal-Wallis rank-sum test followed by the Wilcoxon rank-sum test with continuity correction. Quantitative data are shown as independent data points in box plots indicating the median and interquartile range.

CyTOF data was analysed following a published workflow (*Nowicka et al., 2017*) (update 28 April 2020) with some modifications. Marker expression data was arcsinh transformed with a cofactor of 5. We decided to not normalise the data after visually confirming that the expression pattern distribution did not differ between batches of stain/run days (*Figure 1—figure supplement 2*). Clustering of the single-cell data was performed with FlowSOM (*Van Gassen et al., 2015*) and ConsensusClusterPlus (*Wilkerson and Hayes, 2010*), using the lineage markers labelled as 'type' (*Supplementary file 2*). We clustered each tissue separately and pooled the samples from all donors; we subsampled the data using the 75th percentile of the number of cells per tissue as a threshold, to prevent big samples from driving the clustering (see *Figure 1—figure supplement 3a*, *Figure 4—figure supplement 2a*, *Figure 6—figure supplement 2a*). The metaclusters obtained (20 for the main populations, 12 for the

T cell lineage, and 8 for the NK and B cell lineages) were manually annotated and merged based on visualisation using heatmaps with normalised median marker expression per cluster and dimensionality reduction plots (tSNE and UMAP).

We used a generalised linear mixed model (GLMM) for analysing the differential abundance of cell populations and a linear mixed model (LMM) for the differential expression of 'state' phenotypic markers per cell population. The models were built through the R package *diffcyt*, with a few modifications. The GLMM response variable is the cell counts per cell type (those with more than 100 cells) and sample, and the explanatory variable is the disease group, as a fixed effect. To model the overdispersion in proportion estimates (uncertainty is higher when proportions are calculated from samples with a low total number of cells), we included the sample ID as a random effect. The LMM response variable is the median marker expression of the state markers per cell type (those with more than 100 cells) and sample. Differential expression of markers was only considered in cell populations for which they were biologically meaningful (for example, FoxP3 was only considered in T and B cells), and in which the expression values were higher than 0 in at least a third of the samples. Only cell subsets in which at least one marker was differentially expressed are displayed in the state marker figures. Weights were assigned to each cluster and sample based on the number of cells, to account for differences in uncertainty in the calculation of the medians. We specified the contrasts MS vs. control, MS vs. dementia, and control vs dementia. We corrected the p-values for multiple testing with the Benjamini-Hochberg procedure, with a false discovery rate (FDR) cut-off of 10%; for differential abundance, we corrected for cell population per tissue type, and for differential expression, we corrected for state marker per cell population.

Correlations were performed using the functions *rcorr* and *corr.test*, as implemented in the packages *psych* and *Hmisc*, respectively, using the Benjamini-Hochberg method to adjust p-values. Only patients with samples analysed in all three tissues were used. Significant correlations (adjusted p≤0.1) were visualised in a network plot using Cytoscape. The strength of the correlation was calculated as -log10(adjusted p) and shown as the width of the edges.

## Multiplex immunohistochemistry

Sections of 5-µm-thickness from formalin-fixed paraffin-embedded tissue were cut with a microtome and mounted on SuperFrost Plus slides. Sections were deparaffinised in xylene and rehydrated with a series of graded ethanol (100%, 90%, 80%, and 70% for 2 min). Antigen retrieval was performed in citrate buffer pH 6.0 at 95 °C for 30 min in a water bath. Endogenous peroxidases were blocked using 0.3% $H_2O_2$ in PBS for 15 min. Slides were blocked using 1% bovine serum albumin (BSA) in PBS with 0.05% Tween-20 for 30 min. Sections were then incubated with a single unconjugated primary antibody (*Supplementary file 3*) in 10 x diluted blocking buffer for either 60 min at room temperature or 24 hr at 4 °C.

Slides were washed in PBS-Tween-20 and incubated with Envision +Dual link HRP (Dako, #K4061) for 30 min. Sections were then incubated with the respective Opal fluorochrome (Opal 480, Opal 520, Opal 570, Opal 620, Opal 650, Opal 780) at a 1:250 dilution made in tyramide signal amplification reagent (Akoya, #FP1498) for 60 min. Slides were washed in PBS-Tween-20 and a heating step of citrate buffer pH 6.0 at 98 °C for 30 min in a water bath was performed for primary antibody removal. Afterwards, the tissue was stained by repeating staining cycles in series as described above with a primary antibody removal step in each cycle. Finally, slides were counterstained with DAPI (Invitrogen, #D1306) for 5 min and mounted with Prolong Gold (Invitrogen, #P36930). Slides were imaged using the Vectra 3.0 spectral imaging system (PerkinElmer), with a low magnification scanning at 10 x to get an overview of the slide. To get representative images of the tissue section, a minimum of 2 regions of interest across the fields were chosen and scanned at ×40 magnification. Spectral unmixing was performed using InForm advanced image analysis software (PerkinElmer) and image segmentation was done in Nis Elements (Nikon). In short, nuclei detection was based on the DAPI signal, and a mask was made from nuclei that were directly surrounded by the CD45 signal. This mask was used to further select NK cells based on the NKp46 signal. The NK cell mask was then divided based on granzyme K and granzyme B expression. Cells within the lumen of vessels were excluded manually with the aid of collagen IV staining. All cell counts were then normalised to the total tissue area. If data were normally distributed as assessed by the Shapiro-Wilk test, an ANOVA test or unpaired t-test was used; if the data did not pass

normality, the Kruskal-Wallis test or Mann-Whitney test was used. Staining, imaging, and analysis were performed blinded.

## Acknowledgements

We would like to acknowledge all the donors and their families who made this study possible, as well as the Netherlands Brain Bank, and especially Michiel Kooreman for his support in the collection of the human samples. We thank the Microscopy and Cytometry Core Facility from the Amsterdam UMC for excellent technical support, especially Juan J Garcia Vallejo and Cora Chadick. We appreciate the inspiring discussions with the members of the Molecular Cell Biology and Immunology department, especially Mike de Kok, Jan Verhoeff, and Reina Mebius. We are grateful to Marvin M van Luijn for his valuable input on T-bet+ B cells. Stefanos Prouskas kindly provided information on the lesion location for immunohistochemical validation. We appreciate the support on panel design from Olga Karpus, analysis from Sofie van Gassen, and statistical analysis from Mark van de Wiel. This work was funded by the pilot grant 20–1087 MS from the Dutch MS Research Foundation to SRL, GK, and HEV.

## Additional information

### Funding

| Funder | Grant reference number | Author |
| --- | --- | --- |
| Stichting MS Research | 20-1087 | Sabela Rodríguez-Lorenzo<br>Gijs Kooij<br>Helga E de Vries |

The funders had no role in study design, data collection and interpretation, or the decision to submit the work for publication.

### Author contributions

Sabela Rodríguez-Lorenzo, Conceptualization, Formal analysis, Funding acquisition, Investigation, Methodology, Project administration, Visualization, Writing – original draft, Writing – review and editing; Lynn van Olst, Conceptualization, Formal analysis, Investigation, Methodology, Project administration, Visualization, Writing – original draft, Writing – review and editing; Carla Rodriguez-Mogeda, Investigation, Methodology, Writing – original draft, Writing – review and editing; Alwin Kamermans, Formal analysis, Investigation, Methodology, Visualization, Writing – review and editing; Susanne MA van der Pol, Methodology; Ernesto Rodríguez, Visualization, Writing – review and editing; Gijs Kooij, Funding acquisition; Helga E de Vries, Funding acquisition, Supervision, Writing – review and editing

### Author ORCIDs

Sabela Rodríguez-Lorenzo http://orcid.org/0000-0002-5956-4519
Lynn van Olst http://orcid.org/0000-0001-7569-0470
Carla Rodriguez-Mogeda http://orcid.org/0000-0002-7849-9302
Alwin Kamermans http://orcid.org/0000-0002-3601-395X
Gijs Kooij http://orcid.org/0000-0002-9488-2918
Helga E de Vries http://orcid.org/0000-0001-7904-7124

### Ethics

Human subjects: Post-mortem tissue was obtained from donors by rapid autopsy from the Netherlands Brain Bank and Multiple Sclerosis Society Tissue Bank. All donors or their next of kin provided fully informed consent for autopsy and use of material for research from Netherlands Brain Bank under ethical approval by the Medical Ethics Committee of the Free University Medical Center in Amsterdam (2009/148), project number 1127.

### Decision letter and Author response

Decision letter https://doi.org/10.7554/eLife.73849.sa1
Author response https://doi.org/10.7554/eLife.73849.sa2

## Additional files

### Supplementary files
- Transparent reporting form
- Supplementary file 1. Information of the donors used in this study.
- Supplementary file 2. Antibody panel used for CyTOF.
- Supplementary file 3. Details of antibodies used for multiplex immunohistochemistry.
- Source data 1. Percentage of each annotated cell population out of the total CD45+ cells from the choroid plexus of controls (Con), dementia (AD) and multiple sclerosis (MS) donors. Median scaled intensities of the "state" (phenotype) markers for each cell population.
- Source data 2. Percentage of each annotated cell population out of the total CD45+ cells from the choroid plexus of controls (Con), dementia (AD) and multiple sclerosis (MS) donors. Median scaled intensities of the "state" (phenotype) markers for each cell population.
- Source data 3. Percentage of each annotated cell population out of the total CD45+ cells from the blood of controls (Con), dementia (AD) and multiple sclerosis (MS) donors. Median scaled intensities of the "state" (phenotype) markers for each cell population.

### Data availability
FCS CyTOF files are uploaded to http://flowrepository.org with IDs FR-FCM-Z4JJ (raw) and FR-FCM-Z4JK (normalised). Code used in this analysis is available on the github repository https://github.com/MolecularCellBiologyImmunology/cytof-periventricular-ms, (copy archived at swh:1:rev:70e973c2d935e4ff2cb080d7feef0dd08c23e061).

The following datasets were generated:

| Author(s) | Year | Dataset title | Dataset URL | Database and Identifier |
|---|---|---|---|---|
| Rodríguez-Lorenzo S, Olst L, Rodriguez-Mogeda C, Kamermans A, van der Pol S, Rodríguez E, Kooij G, de Vries HE | 2022 | Investigating the periventricular immune landscape in MS using CyTOF | http://flowrepository.org/id/FR-FCM-Z4JJ | FlowRepository, FR-FCM-Z4JJ |
| Rodríguez-Lorenzo S, Olst L, Rodriguez-Mogeda C, Kamermans A, van der Pol S, Rodríguez E, Kooij G, de Vries HE | 2022 | Investigating the periventricular immune landscape in MS using CyTOF | https://flowrepository.org/id/FR-FCM-Z4JK | FlowRepository, FR-FCM-Z4JK |

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
