## [Editor Report]

This is a well-written, well-illustrated and well-conducted study of the immune cell landscape of multiple sclerosis (MS) tissue, with a particular focus on the periventricular region (septum) and choroid plexus, using single-cell mass cytometry (CyTOF). Overall the work is an impressive analysis of an understudied cell type in MS and represents an important resource.

---

## [Decision Letter]

**Decision letter after peer review:**

Thank you for submitting your article "Single-cell profiling reveals periventricular CD56^bright^ NK cell accumulation in multiple sclerosis" for consideration by *eLife*. Your article has been reviewed by 3 peer reviewers, one of whom is a member of our Board of Reviewing Editors, and the evaluation has been overseen Satyajit Rath as the Senior Editor. The reviewers have opted to remain anonymous.

*Essential revisions:*

This is a well-written, well-illustrated and well-conducted study of the immune cell landscape of MS tissue, with a particular focus on the periventricular region (septum) and choroid plexus, using single cell mass cytometry (CyTOF). Overall the work is an impressive analysis of an understudied cell-type in MS, and represents an important finding. The paper is well presented and the figures very clear.

However, the manuscript is highly descriptive and the depth and limitations of the Cytof leaves the reader without a clear idea of what these cells could be doing. A major weakness of the study is that is underpowered and thus not clear how robust or representative these findings are in MS given the heterogeneity of the disease and also potential differences in Sex, Age and lack of healthy controls. (AD samples labelled as control.) While the patterns found are necessarily descriptive as yet, these are extensive hard-to-get data that well deserve publication the following key issues and concerns are addressed first.

1. The depth and limitations of the Cytof leaves the reader without a clear idea of what these cells could be doing. At minimum more immunohistochemical and smFish or in situ hybridization to validate key findings (using the markers identified by CyTOF) and add to the spatial relationships of Nk Cells with other border and brain cells would be informative.

2. Details on samples and controls should be more clearly communicated in the text and legends as well as the caveats and limitations of the study in the discussion. See specific comments below related to potential sex differences and gender distributions. Have the gender ratio and the age distribution been excluded in interpreting the data for the NK cell populations?

3. The authors reference the initiation of immune lesions in early MS. However, the work itself addresses end-stage MS situations, which is quite possibly an entirely different landscape altogether, and may not be informative about MS initiation. Moreover, the manuscript makes far too many speculations about possible cell trafficking between compartments than is justified by a cross-section study.

4. Are the NK cells actually in the parenchyma and interacting with other cells (eg. microglia) of the lesion. If the authors have this tissue and antibodies to do that, this would add to the study.

5. The relative prominence of leukocytes in the septum tissue is a missing analysis for completeness of the CyTOF data. (See suggestion below.)

6. The dendritic cell component in the 'myeloid APC' group could use further characterisation to the extent feasible, given potential NK cell-DC crosstalk, especially since there appears to be a selective absence of T-bet-expressing potentially cytotoxic NK cells.

7. Please address: Do the CD56bright NK cells show a normal distribution for the markers examined in figure 3d, or are there subsets of NK cells revealed when examining these markers? Do all CD56bright NK cells show similar T-bet expression (with the per cell T-bet level lower in the MS group), or are there subsets? And what is the GATA-3 status on these NK cell populations (given the GATA-3 prominence in cytokine-producing ILC2-like cells)?

*Reviewer #3 (Recommendations for the authors):*

While the patterns found are necessarily descriptive as yet, these are extensive hard-to-get data that well deserve publication, although it would be important to have some issues and concerns addressed first.

1. The non-neurological control group shows a very different gender distribution than the other two groups, the three groups show differences in age ranges (supplementary figure 1), and there are clearly gender-based and age-based differences in at least some of the cell subsets examined (supplementary figure 2). Have the gender ratio and the age distribution been excluded in interpreting the data for the NK cell populations, which is the most striking finding emerging?

2. The relative prominence of leukocytes in the septum tissue is a missing analysis for completeness of the CyTOF data. This could be done either as yields of CD45-expressing cells per quantum of tissue, or as CD45 transcript levels in total tissue RNA. Further from this, some estimate of the prominence of each immune cell subset identified per quantum of tissue would be very informative.

3. The dendritic cell component in the 'myeloid APC' group could use further characterisation to the extent feasible, given potential NK cell-DC crosstalk, especially since there appears to be a selective absence of T-bet-expressing potentially cytotoxic NK cells.

4. Does the relative prominence of NK cells show any correlations with that of CD4 (or CD8) T cells?

5. A previous report, cited by the authors (their reference 46), shows that CD56bright NK cells are specifically increased in the CSF in MS, and that this is accompanied by an increase in NK T cells. Is this the case in the septum and/or choroid plexus in the present dataset (NK T cells do not seem to have been clearly identified in the present analyses)?

6. Is T-bet expression altered in the MS group in CD4/CD8 effector/memory T cells as well? (It IS lower both in γ-δ T cells and NK cells in this group.)

7. Do the CD56bright NK cells show a normal distribution for the markers examined in figure 3d, or are there subsets of NK cells revealed when examining these markers? A sub-query of this general question is; do all CD56bright NK cells show similar T-bet expression (with the per cell T-bet level lower in the MS group), or are there subsets? And what is the GATA-3 status on these NK cell populations (given the GATA-3 prominence in cytokine-producing ILC2-like cells)?

8. The immunohistochemistry data provide a small but interesting indication that location in the lesion makes a difference; its appears that the CD56bright/CD56dim (or at least, the granzyme K/granzyme B) NK cell ratio may be quite different for MS lesion centres versus borders. Given the potential functional differences in these two subsets, this might be worth confirming and discussing if confirmed.

9. The sentence in the 'results' section, that 'Matching the CyTOF data, GrB+ NK cells were more abundant than GrK^+^ cells', is puzzling; what are the data that this sentence refers to? Assuming that it refers to findings in choroid plexus immunohistochemistry in figure 8d, the data do not actually seem to show what the sentence says, nor do they seem to support the CyToF findings in the choroid plexus. This would make sense if, in fact, the data immunohistochemistry data exclude intravascular NK cells in the choroid plexus, but this is not described. A clarification would be useful.

---

## [Author Response]

Reviewer #3 (Recommendations for the authors):While the patterns found are necessarily descriptive as yet, these are extensive hard-to-get data that well deserve publication, although it would be important to have some issues and concerns addressed first.1. The non-neurological control group shows a very different gender distribution than the other two groups, the three groups show differences in age ranges (supplementary figure 1), and there are clearly gender-based and age-based differences in at least some of the cell subsets examined (supplementary figure 2). Have the gender ratio and the age distribution been excluded in interpreting the data for the NK cell populations, which is the most striking finding emerging?

We acknowledge that it was not possible to perform age/sex matching between all the experimental groups from which fresh brain tissue was taken for CyTOF phenotyping. However, the sex ratio in the non-neurological control and MS groups is the same (Figure 1 —figure supplement 1), and these two groups are the main base for our findings of an accumulation of CD56^bright^ NK cells in the MS periventricular brain. Regarding the age differences, the MS group is younger than the controls. Given the known increase in CD8^+^ T cells with ageing (Smolders et al., 2018), our finding of higher numbers of this subset in the younger MS group is only strengthened, as well as in line with previous literature (Machado-Santos et al., 2018). It has been reported before that CD56^bright^ NK cells decline in peripheral blood with increasing age (Chidrawar et al., 2006). However, we validated the accumulation of CD56^bright^ NK cells in MS with immunohistochemistry in an independent cohort. This cohort presented a similar mean age between controls (65.8y) and MS cases (62.5y) (Wilcoxon rank-sum test with continuity correction, *P*-value = 0.41) and the sex ratio was also similar with a control f:m ratio of 0.667 and MS f:m ratio of 0.75). We have now highlighted the characteristics (age and sex) of the IHC validation cohort in the results.

Finally, to confirm that CD56^bright^ NK cells accumulate in periventricular brain regions in MS donors, we used multiplex immunohistochemistry in an independent cohort (Supplementary File 1), wherein MS and control groups were age-matched (Wilcoxon rank sum test with continuity correction, p-value = 0.41) and had a similar female:male ratio (0.667 in controls and 0.75 in MS).

Smolders J, Heutinck KM, Fransen NL, Remmerswaal EB, Hombrink P, Ten Berge IJ, van Lier RA, Huitinga I, Hamann J (2018) Tissue-resident memory T cells populate the human brain. Nature communications 9:1-14

Machado-Santos J, Saji E, Tröscher AR, Paunovic M, Liblau R, Gabriely G, Bien CG, Bauer J, Lassmann H (2018) The compartmentalized inflammatory response in the multiple sclerosis brain is composed of tissue-resident CD8^+^ T lymphocytes and B cells. Brain 141:2066-2082

Chidrawar, Shivani M., et al. "Ageing is associated with a decline in peripheral blood CD56 bright NK cells." Immunity and Ageing 3.1 (2006): 1-8.

2. The relative prominence of leukocytes in the septum tissue is a missing analysis for completeness of the CyTOF data. This could be done either as yields of CD45-expressing cells per quantum of tissue, or as CD45 transcript levels in total tissue RNA. Further from this, some estimate of the prominence of each immune cell subset identified per quantum of tissue would be very informative.

We appreciate the interest in absolute immune cell numbers. Due to technical limitations, we do not have the sample weight for all samples to calculate the yield (cells/mg of tissue). Of the samples that had this information, we included the yield of total isolated immune cells (as counted after isolation from the Percoll gradient) per mg of tissue (Author response table 1; Author response image 1). However, we want to stress that the total yield of cells, especially in human post-mortem tissue, is influenced by many technical and biological factors and we, therefore, find that the percentage of each cell subset from the total amount of immune cells more reliable to work with.

**Author response table 1. sa2table1:** Yield of isolated live immune cells per mg of septal tissue.

**Donor**	**Yield (cells/mg)**	**Condition**
2018-120	4782.3	Control
2018-123	1666.2	Control
2019-054	3231.4	Control
2019-079	3596.2	Control
2020-003	4861.1	Control
2020-028	1110.4	Control
2019-030	3286.3	Dementia
2020-013	3607.5	Dementia
2019-016	3052.9	MS
2019-027	2183.0	MS
2019-031	6232.0	MS
2019-069	3175.9	MS
2019-107	2288.9	MS
2019-109	1345.3	MS
2019-110	2846.6	MS

**Author response image 1. sa2fig1:** Absolute live immune cell numbers per mg of septum.

3. The dendritic cell component in the 'myeloid APC' group could use further characterisation to the extent feasible, given potential NK cell-DC crosstalk, especially since there appears to be a selective absence of T-bet-expressing potentially cytotoxic NK cells.

We studied in more detail the subpopulations of CD56bright NK cells and myeloid cells as seen in Author response image 2. Regarding CD56bright NK cells, we could sub-cluster this population into two groups, which interestingly were characterized as T-bet- and T-bet+ (Essential Revisions Figure 3). As the reviewer points out, we observed a decrease of T-bet+ CD56bright NK cells in MS (Figure 3 —figure supplement 2 c), which fits the decrease in expression observed in Figure 3a in the manuscript. This absence of T-bet on CD56bright NK cells could indicate a loss of cytotoxicity (Harmon et al., 2016) or developmental immaturity (Collins et al., 2017) which is also associated with a hampered ability to kill target cells in comparison to more mature subsets (Oei et al., 2018). We have added this data to Figure 3 —figure supplement 2 and discuss these new data in the results:

“Further sub-clustering of CD56bright NK cells revealed the existence of a Tbet+ and a Tbet- subset (Figure 3 —figure supplement 2). Tbet- NK cells expressed higher levels of molecules that were increased in the general CD56bright NK cluster of MS patients (Figure 3a), and were the dominant CD56bright cell type in MS but not in control or dementia cases (Figure 3 —figure supplement 2). Overall, this suggests that specifically Tbet- CD56bright NK cells are increased in the septum of MS donors.”

**Author response image 2. sa2fig2:** CD56^bright^ NK cells and myeloid cells subpopulations in septum. (a) UMAPs displaying pre-gated CD56^bright^ NK subpopulation in the septum. The cells are coloured according to PARC-guided clustering. (b) Median marker intensities of pre-selected markers across CD56^bright^ NK subpopulation of the septum. Horizontal grey bars show the percentage out of the total cells. The colour in the heatmap represents the median of the arcsinh, 0-1 transformed marker expression calculated over cells from all the samples. (c) Stacked bargraph show the fraction of T-bet^-^ and T-bet^+^ clusters measured by ANOVA. (d) UMAPs displaying pre-gated myeloid subpopulation in the septum. The cells are coloured according to PARC-guided clustering. (e) Median marker intensities of pre-selected markers across myeloid subpopulation of the septum. Horizontal grey bars show the percentage out of the total cells. The colour in the heatmap represents the median of the arcsinh, 0-1 transformed marker expression calculated over cells from all the samples. (f) Stacked bargraph show the fraction of each of the three clusters. ** P < 0.01. Con: controls; Dem: dementia; MS: multiple sclerosis.

We could detect 3 subpopulations of myeloid cells (80 % were CD11chigh CD49d+ CD206-, 12.8 % were CD11clow CD49d+ CD206+; and 7.3 % were CD11chigh CD49d- CD206-) (Author response image 2). but we cannot clearly identify them as the main known myeloid cells populations with our markers (Author response image 2)). Cluster 1 and 3 could be dendritic cells and cluster 2 macrophages. Nevertheless, we did not observe differences between Dementia, MS, and controls regarding these 3 myeloid subsets (Figure 3 —figure supplement 2). From this data, we cannot extrapolate whether NK cells and DCs are having potential crosstalk. We have added these data to Figure 1 —figure supplement 4 and discussed it in the Results section of our manuscript “APCs of myeloid origin consisted of 3 subsets (80 % were CD11chigh CD49d+ CD206-, 12.8 % were CD11clow CD49d+ CD206+; and 7.3 % were CD11chigh CD49d- CD206-) (Figure 1 —figure supplement 4).”

4. Does the relative prominence of NK cells show any correlations with that of CD4 (or CD8) T cells?

We thank the reviewer for this question which the answer might shed some light on the interaction of NK cells with other key lymphocytes in MS pathology. We have not included this in the initial version of our paper, where we focused on inter-tissue correlations and did not further elaborate on correlations within each tissue. We did not find a correlation between septum-derived NK and T cells (both CD4^+^ and CD8^+^). For extra information, Author response image 3 are the correlation plots in the controls and MS groups separately

**Author response image 3. sa2fig3:** Spearman correlation matrix of the proportion of the main immune cell populations among the studied tissues in the control and MS samples, separately. Positive correlations are displayed in red and negative correlations in blue. The size of the circle is proportional to the correlation coefficients. P-values adjusted with Benjamini-Hochberg method, excluding intra-tissue correlations. * adjusted P < 0.1.

5. A previous report, cited by the authors (their reference 46), shows that CD56bright NK cells are specifically increased in the CSF in MS, and that this is accompanied by an increase in NK T cells. Is this the case in the septum and/or choroid plexus in the present dataset (NK T cells do not seem to have been clearly identified in the present analyses)?

While we detected NKT cells in the blood (Figure 6), our study was not enough powered to identify NKT cells in the CNS samples (Figures 1 and 4). Therefore, we cannot comment on the possible involvement of CP or septum-derived NKT cells in MS.

6. Is T-bet expression altered in the MS group in CD4/CD8 effector/memory T cells as well? (It IS lower both in γ-δ T cells and NK cells in this group.)

Due to a large amount of data, out of all the “state” markers, we display only those which show significant differential expression between the studied groups. In Figure 2c, we show that γδ T cells in MS donors express lower levels of T-bet than those in controls. This is not the case for other subsets, such as effector/memory, of CD4^+^ or CD8^+^ T cells.

We noticed that some figure legends mention “selected markers”, which is misleading and we have removed it. For example, in Figure 2c:

“Boxplots show median expression in γδ T cells of selected markers showing differential expression between the MS and NNC samples”,

We have removed the word “selected”. New version:

“Boxplots show median expression in γδ T cells of markers showing differential expression between the MS and control samples.”

For further clarification, we added a sentence in the CyTOF data processing and statistical analysis section of the Methods:

“Only cell subsets in which at least one marker was differentially expressed are displayed in the state marker figures.”

7. Do the CD56bright NK cells show a normal distribution for the markers examined in figure 3d, or are there subsets of NK cells revealed when examining these markers? A sub-query of this general question is; do all CD56bright NK cells show similar T-bet expression (with the per cell T-bet level lower in the MS group), or are there subsets? And what is the GATA-3 status on these NK cell populations (given the GATA-3 prominence in cytokine-producing ILC2-like cells)?

We studied in more detail the subpopulations of CD56bright NK cells as seen in Author response image 2 Regarding CD56bright NK cells, we could sub-cluster this population into two groups, which interestingly were characterized as T-bet- and T-bet. As the reviewer points out, we observed a decrease of T-bet+ CD56bright NK cells in MS (Figure 3 —figure supplement 2 c), which fits the decrease in expression observed in Figure 3a in the manuscript. This absence of T-bet on CD56bright NK cells could indicate a loss of cytotoxicity (Harmon et al., 2016) or developmental immaturity (Collins et al., 2017) which is also associated with a hampered ability to kill target cells in comparison to more mature subsets (Oei et al., 2018). We have added this data to Figure 3 —figure supplement 2 and discuss these new data in the results “Further sub-clustering of CD56bright NK cells revealed the existence of a Tbet+ and a Tbet- subset (Figure 3 —figure supplement 2). Tbet- NK cells expressed higher levels of molecules that were increased in the general CD56bright NK cluster of MS patients (Figure 3a), and were the dominant CD56bright cell type in MS but not in control or dementia cases (Figure 3 —figure supplement 2). Overall, this suggests that specifically Tbet- CD56bright NK cells are increased in the septum of MS donors.

The subcluster of T-bet+ CD56bright NK cells showed a dim expression of GATA3, which could suggest a more ILC2-like cell, but it is not possible to further characterize this population.

8. The immunohistochemistry data provide a small but interesting indication that location in the lesion makes a difference; its appears that the CD56bright/CD56dim (or at least, the granzyme K/granzyme B) NK cell ratio may be quite different for MS lesion centres versus borders. Given the potential functional differences in these two subsets, this might be worth confirming and discussing if confirmed.

We want to thank the reviewer for this thoughtful comment on the different ratios of GrK^+^/GrB+ NK cells in the border and center of the lesions, as observed by immunohistochemistry. CD56bright NK cells (measured with the surrogate marker GrK) abound in the rim of the lesions, where the highest activity is present in terms of immune cell activation and demyelination. We can speculate that this is the ideal location for CD56bright NK cells to exert immunosuppression and try to contain the damage. The center of MS lesions is characteristically hypocellular, and this is reflected in the number of NK cells present. We have discussed these differences further in the Results section manuscript. “GrK^+^ NK cells accumulated in both the border and the centre of the lesion compared to NAWM, and were more abundant than GrB+ NK cells (Figure 8b-c). The increased presence of GrK^+^ NK cells in the MS lesion confirms our earlier CyTOF data where the MS septum contained more CD56bright than CD56dim NK cells, while CD56dim NK cells were the dominant NK subset in the septum derived from controls and donors with dementia (Figure 3c). And in the discussion: “Putative CD56bright NK cells abound in the rim of the lesions, where the highest activity is present in terms of immune cell activation and demyelination. The center of MS lesions is characteristically hypocellular, and this is reflected in the number of NK cells present.”

9. The sentence in the 'results' section, that 'Matching the CyTOF data, GrB+ NK cells were more abundant than GrK^+^ cells', is puzzling; what are the data that this sentence refers to? Assuming that it refers to findings in choroid plexus immunohistochemistry in figure 8d, the data do not actually seem to show what the sentence says, nor do they seem to support the CyToF findings in the choroid plexus. This would make sense if, in fact, the data immunohistochemistry data exclude intravascular NK cells in the choroid plexus, but this is not described. A clarification would be useful.

First, we apologize for the typo in our sentence; it should read: “Matching the CyTOF data, GrK^+^ NK cells were more abundant than GrB+ cells”. We have corrected and rephrased it for further clarity into: “GrK^+^ NK cells accumulated in both the border and the centre of the lesion compared to NAWM, and were more abundant than GrB+ NK cells (Figure 8b-c). The increased presence of GrK^+^ NK cells in the MS lesion confirms our earlier CyTOF data where the MS septum contained more CD56bright than CD56dim NK cells, while CD56dim NK cells were the dominant NK subset in the septum derived from controls and donors with dementia (Figure 3c).